# A population study of *Toxoplasma gondii* in the Amazon region expands current knowledge of the genetic diversity in South America

Solange M. Gennari[1], Hilda F. J. Pena[1,2]*, Herbert S. Soares[1,2], Antonio H. H. Minervino[3], Francisco F. V. de Assis[3], Bruna F. Alves[1¤], Solange Oliveira[1], Juliana Aizawa[1], Ricardo A. Dias[1], Chunlei Su[4]

**1** Department of Preventive Veterinary Medicine and Animal Health, Faculty of Veterinary Medicine and Zootechny, University of São Paulo, São Paulo, São Paulo, Brazil, **2** Postgraduate Program in One Health, University of Santo Amaro, São Paulo, São Paulo, Brazil, **3** Laboratory of Animal Health, Federal University of Western Pará, Santarém, Pará, Brazil, **4** Department of Microbiology, University of Tennessee, Knoxville, Tennessee, United States of America

¤ Current address: Department of Veterinary Medicine, Faculdade Empresarial de Chapecó, UCEFF, Chapecó, Santa Catarina, Brazil

* hfpena@usp.br

## Abstract

Previous studies have reported high diversity between and within populations of *Toxoplasma gondii* in South America. In the present study, isolates of *T. gondii* from chickens were obtained from the Amazon region. Adult free-range chickens were acquired from 29 municipalities in the Brazilian Amazon region that included Acre (n = 9 municipalities), Amapá (n = 6), Amazonas (n = 6), Pará (n = 6), and Roraima (n = 2) states and from two municipalities in Peru, three in Bolivia, one in Guyana, and one in Venezuela. Heart, brain, and blood samples were collected from 401 chickens. Anti-*T. gondii* serum antibodies were detected in 273 (68.1%) chickens using the Modified Agglutination Test (MAT ≥ 5), and bioassays in mice were performed using 220 birds. Isolates were obtained from 116 (52.7%) chickens with antibody titers ≥ 20. Of these isolates, 93 (84.5%) led to acute sickness in more than 50% of the infected mice within 30 days post-inoculation. The 116 isolates were genotyped using multilocus nested polymerase chain reaction-restriction fragment length polymorphism (Mn-nPCR-RFLP) with 12 markers and 15 microsatellite (MS) markers. PCR-RFLP analysis revealed 42 genotypes from the 116 isolates. Of these, 20 (46.51%) genotypes are described for the first time. The most abundant genotype was ToxoDB PCR-RFLP genotype #7 with 40 isolates. A total of 83 genotypes were observed from the 116 isolates by MS analysis. The phylogenetic network constructed of *T. gondii* genotypes from current and previously reported isolates, using PCR-RFLP data, revealed five groups with clear indication of geographical separation of *T. gondii* population in the Amazon region versus the Southeastern region of Brazil. Such spatial diversity was also observed within the Amazon region. This study expands our knowledge of *T. gondii* population in South America and emphasizes the importance of genetic diversity and high mouse-virulence of the parasite in the Amazon region.

**Data availability statement:** All relevant data are included in the manuscript and its Supporting Information files.

**Funding:** This work was supported by funds from Fundação de Amparo à Pesquisa do Estado de São Paulo (FAPESP, https://fapesp.br/, project grant 2015/11530-6 given to SMG and project grant 2018/26071-5 given to HFJP), from Conselho Nacional de Desenvolvimento Científico e Tecnológico (CNPq, https://www.gov.br/cnpq/pt-br, given to SMG and HSS as productivity scholarship and to FFVA as undergratuate research scholarship) and from Coordenação de Aperfeiçoamento de Pessoal de Nível Superior-Brasil (CAPES, https://www.gov.br/capes/pt-br, Finance code 001, given to HFJP). The funders had no role in study design, data collection and analysis, decision to publish or preparation of the manuscript.

**Competing interests:** The authors have declared that no competing interests exist.

## Author summary

In the present study, we obtained isolates of *Toxoplasma gondii* from chickens from the Amazon region. Adult free-range chickens were acquired from 29 municipalities in the Brazilian Amazon region including Acre (n = 9), Amapá (n = 6), Amazonas (n = 6), Pará (n = 6), and Roraima (n = 2) states and from two municipalities in Peru, three in Bolivia, one in Guyana, and one in Venezuela. Heart, brain, and blood samples were collected from 401 chickens. We detected anti-*T. gondii* serum antibodies in 273 (68.1%) chickens using the Modified Agglutination Test (MAT ≥ 5), and bioassays in mice were performed using 220 birds. We obtained 116 (52.7%) isolates from chickens with antibody titers ≥ 20. Of these isolates, 93 (84.5%) were considered virulent to mice. We genetically analyzed the 116 isolates using multilocus nested polymerase chain reaction-restriction fragment length polymorphism (Mn-nPCR-RFLP) and microsatellites (MS). PCR-RFLP analysis revealed 42 genetic types and MS analysis revealed 83 genetic types. When we analyzed and compared *T. gondii* from current and previously reported isolates, using PCR-RFLP data, we observed a clear indication of geographical separation of *T. gondii* population in the Amazon region versus the Southeastern region of Brazil. This study emphasizes the importance of genetic diversity and high mouse-virulence of the parasite in the Amazon region.

## Introduction

*Toxoplasma gondii* infection is prevalent in animals and humans worldwide [1], and felids are the definitive hosts in which sexual reproduction leads to the production of millions of highly resistant oocysts [2].

There is extensive interest in the molecular investigation of *T. gondii* from animals and humans to assess the correlation between the parasite genotypes and their biological phenotypes, and to trace the sources of infection and the routes of transmission. Molecular epidemiological techniques are of great importance in these studies. Previous studies revealed limited genetic diversity in *T. gondii* from the Northern Hemisphere and other regions of the world. Among the different regions, Type II (PCR-RFLP genotypes #1 and #3) and Type III (PCR-RFLP genotype #2) predominate in Europe. Type II (#1 and #3), Type III (#2), and Type 12 (#4 and #5) predominate in North America. Type II variant (#3), Type III (#2), Africa 1 (#6, Type BrI), Africa 3 (#203) and Africa 4 (#20) are prevalent in Africa. Type I (#10), Type II (#1 and #3), Type III (#2), and the regional Chinese 1 (#9) dominate in Asia, particularly in China [3–6].

However, studies examining isolates from South America, particularly from Brazil and Colombia, have demonstrated that they are biologically and genetically different from those from North America and Europe. These isolates have high genetic diversity and pathogenicity in murine models [6–10] that may be linked to more severe forms of toxoplasmosis [11–13].

*Toxoplasma gondii* transmission in the Amazon region is a complex which involves wild animals, humans living close to the rainforest, and non-archetypal strains [14]. It was demonstrated that *T. gondii* isolates from French Guiana can cause very serious clinical manifestations in humans [15,16]. Fatal clinical cases of toxoplasmosis in healthy humans had occurred in the region after consumption of wild animal meat [17]. Cases of severe acute toxoplasmosis have been reported in Peru [18] and Brazil [19,20]. In Brazilian Amazon riverside communities, Vitaliano et al. [21] found a seroprevalence of 56.7% for *T. gondii* antibodies and five

different isolates were obtained in chicken bioassay. All these findings highlighted the importance of studying the epidemiology of toxoplasmosis in the region.

Currently, there is a lack of data regarding *T. gondii* diversity in the Amazonian rainforest. Vitaliano et al. [22] conducted a phylogenetic study on *T. gondii* isolates from wild animals in Brazil (including samples from the Amazon region) and observed the presence of a different phylogenetic branch. Although the data did not show the existence of a unique Amazon cluster, the results suggest that this branch may be dominant in the area.

The aim of this study was to improve our understanding of the circulating *T. gondii* genotypes in the Amazon region and the geographic distribution of this parasite.

## Materials and methods

### Ethics statement

All procedures were conducted according to the animal protocols approved by the Ethics Committee of the Faculty of Veterinary Medicine and Zootechny, University of São Paulo, São Paulo, SP, Brazil (CEUA/FMVZ/2762070417), following the National Council Guide for the Care and Use of Laboratory Animals.

Genetic heritage access is registered at the System for the Management of Genetic Heritage and Associated Traditional Knowledge (SisGen/ AD4A8A1).

### Chicken sampling

Adult free-range chickens (*Gallus gallus domesticus*) were acquired by convenience from rural and urban areas of 29 municipalities located in the Brazilian Amazon region from 2015-2017 (Fig 1). Regions sampled included the states of Acre (n = 9), Amazonas (n = 6), Amapá (n = 6), Pará (n = 6), and Roraima (n = 2) and from seven municipalities located in the borders of Brazil in Bolivia (n = 3), Peru (n = 2), Guyana (n = 1), and Venezuela (n = 1). In total, 179 small properties were visited. The presence of cats was investigated when possible. Samples of hearts, heads (to extract the brain), and blood from 401 chickens were obtained and sent, by air, in foam boxes with recyclable ice packs, to the Faculty of Veterinary Medicine and Zootechny at the University of São Paulo in São Paulo state to be processed. Detailed information regarding the chicken samples and localities is provided in S1 Table.

### *Toxoplasma gondii* serodiagnosis

The chicken sera were tested for anti-*T. gondii* antibodies using the Modified Agglutination Test (MAT) as described by Dubey and Desmonts [23]. The MAT has been previously validated as a serological test for *T. gondii* diagnostic in chickens [24,25]. A cutoff of 1:5 was used for chicken sera [24]. Initially, chicken sera were screened for three dilutions (1:5, 1:10, and 1:20) and further tested in serial 2-fold dilutions until the final positive dilution is identified.

### Bioassay in mice and *Toxoplasma gondii* isolation

The brain and heart homogenates of seropositive chickens with MAT titers ≥ 10, with some exceptions, were individually bioassayed in Swiss Webster mice for *T. gondii* isolation using 3 to 5 mice per chicken. Bioassays from chickens with serologic titer 5 were performed only partially, if there were mice available for inoculation.

The tissues were homogenized, digested in acidic pepsin, neutralized with sodium bicarbonate, and subcutaneously inoculated into mice [26]. Mice were observed twice daily for infection signs associated with *T. gondii* until 45 days after inoculation and were ethically euthanized according to a score for coat condition (no cumulative score) and behavior

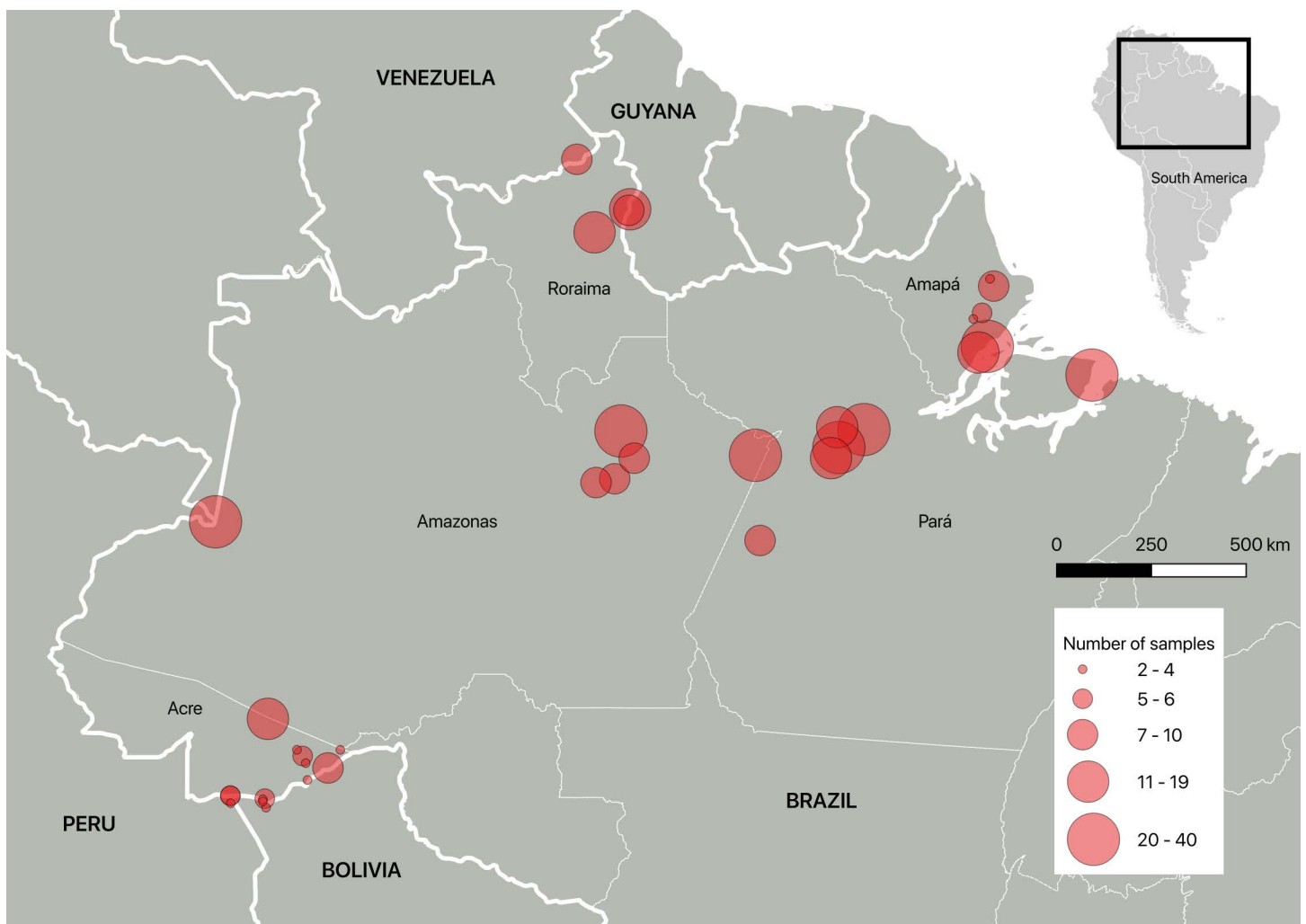

**Fig 1. Free-range chicken sampling localities in the Amazon region.** Chickens were sampled from five Brazilian states (Acre, Amapá, Amazonas, Pará and Roraima) and four countries bordering Brazil (Bolivia, Guyana, Peru and Venezuela). Sample size is represented by the relative size of circle on the map. Data were plotted using QGIS software (basemap from https://www.ibge.gov.br/geociencias/organizacao-do-territorio/malhas-territoriais/15774-malhas.html).

(cumulative score) [27]. The scores included glossy coat (0), ruffled (1), and stiff (2) and active (0), hunched (1), reluctant to move (1), and staggering (1). The animals were euthanized with a total score of "3" for two consecutive days. All mice were intramuscularly sedated with a combination of ketamine (100 mg/kg) and xylazine (10 mg/kg) and euthanized using isoflurane inhalation in a sealed box. The inoculated mice were considered infected with the parasite when tachyzoites or tissue cysts were present in the lungs and brain, respectively [1].

Positive mouse tissues (lungs and/or brain) were macerated separately and homogenized with 1.5 mL of sterile 0.85% NaCl solution. One portion of the homogenate (~500 μL) was stored in microtubes at −70 °C until use in molecular studies, and the other portion was cryopreserved in liquid nitrogen using a solution of 10% BFS + 5% DMSO in RPMI medium. Mice without clinical symptoms of infection were terminated at day 45 post inoculation (p.i.) if seronegative and at day 60 p.i. if seropositive by MAT. A titer of 25 or higher was considered positive in mice [28].

## Genotypic characterization of *Toxoplasma gondii*

For DNA extraction, the macerates of lungs or brain from positive mice (two mice per isolate when there were at least two infected mice) were thawed, and 250-μl aliquots were washed in Tris-EDTA buffer (Tris-HCl 10 mM, EDTA 1 mM) by centrifugation at $12,000 \times g$ for 5 min. Then, the DNeasy Blood & Tissue commercial kit (Qiagen Inc., USA) was used according to the manufacturer's protocol.

Multilocus nested-PCR-RFLP genotyping (Mn-nPCR-RFLP) was performed using the markers SAG1, 5′3′SAG2, alt. SAG2, SAG3, BTUB, GRA6, c22-8, c29-2, L358, PK1, Apico [29] and CS3 [9]. The reference strains RH (Type I), PTG (Type II), and CTG (Type III) and the non-archetypal strains TgCgCa1 (Cougar), MAS, and TgCatBr5 were used as positive controls in all reactions. The PCR-RFLP genotypes of the isolates were compared and classified according to the genotypes available in the ToxoDB database (http:/toxodb.org/toxo/), in major reviewer papers [6,30], in the Dr. Chunlei Su laboratory webpage (https://faculty.utk.edu/Chunlei.Su), and in recent publications.

Additionally, isolates were genotyped using eight microsatellite typing markers (TUB2, W35, TgMA, B18, B17, M33, IV.1, and XI.1) and seven fingerprinting markers (N60, N82, AA, N61, N83, M48, and M102) with primers, fluorophores, a reaction mixture, and cycling conditions as previously described [31]. The sizes of the alleles in base pairs were estimated using the GeneMapper 4.1 software program (Applied Biosystems). The reference strain PTG (Type II) was used as the positive control. The results obtained in this study were compared to MS results from Brazilian *T. gondii* strains that were previously published (S2 Table).

## Data analysis

A composite dataset of multilocus PCR-RFLP genotyping was analyzed using SplitsTree4 (v.4.16.1) software [32,33] to reveal the phylogenetic relationships of all parasites isolates from the present study with other isolates previously described in the Northern and Southeastern regions of Brazil (S3 Table). The results are presented as a reticulated network that is preferred over the traditional bifurcating phylogenetic tree for describing complex relationships in population biology [34]. Detailed information regarding the strains used in this analysis is provided in S3 Table.

The genetic diversity among *T. gondii* populations described in the present study and among these populations and populations previously described in the Southeastern region of Brazil (S3 Table) was assessed and compared using Simpson's diversity index (*D*) [35].

Only strains with complete genotypes as determined by PCR-RFLP markers were included in the comparison studies.

Population analysis was conducted to compare genetic diversity of *T. gondii* isolates following the previously described method [10].

## Statistical analysis

All statistical tests were performed using R Statistical Software (version 4.1.2; R Core Team, 2021) with a confidence level of 5%. To test the normality of the distributions of quantitative data, the Kolmogorov-Smirnov test was used through the *ks.test* () function of R. To compare the medians of two samples, the Mann-Whitney test was used through the *wilcoxon.test* () function of R. Finally, to test the association between two qualitative variables, the chi-square (or Fischer as post-hoc) test was used through the *chisq.test* () function of R.

## Results

### *Toxoplasma gondii* serodiagnosis and isolation

A total of 401 chickens were obtained from 179 properties in 36 municipalities in the Amazon region, 273 (68.1%) chickens had MAT ≥ 5 and were considered positive with *T. gondii* infection. Of the 179 properties visited, 147 (82.1%) had seropositive chickens, and 32 (17.9%) were negative in *T. gondii* infection. Information regarding the presence of cats was obtained from 165 properties (355 chickens were collected). Of the 165 properties, the absence of cats was reported in 42 (25.4%), and 123 (74.6%) reported possessing one to 11 cats and/or observing cats or wild felids from the neighborhood. There was no statistical association between the presence of cats and chicken seropositivity (p = 0.26) and no statistical association was observed between presence of cats and serologic status of the properties (p = 0.85).

Mice bioassays were performed for 220 chicken samples and 116 (52.7%) isolates were obtained from chickens with MAT titers of 20 or higher (103 isolates from Brazil and 13 isolates from the other South American countries studied). No isolates were obtained from 37 chickens with MAT titers 5 and 10. Table 1 summarized the origins of the chickens, the number of birds that were positive for anti-*T. gondii* antibodies, and the number of isolates obtained.

The number of isolates obtained per antibody titer is summarized in Table 2. There was a statistically significant association between the serologic titer and *T. gondii* isolation (the median titer with isolation, n = 240, was higher than the median titer without isolation, n = 40, p < 0.00001). Isolates were obtained from 27 of 36 municipalities (22 Brazilian municipalities and five from other countries investigated). Linear regression analysis of log (% of success) versus MAT titers from 5 to 1,280 revealed Y = 39.6X-28, whereas Y is the success rate in percentage, X is the log (titers). The adjusted R square = 0.89, with the P value <0.0001. MAT titers from 2,560 to 10,240 are excluded from analysis due to small sample sizes (Table 2).

Of these 116 isolates, 98 (84.5%) caused acute sickness in more than 50% of the infected mice and resulted in these mice being euthanized within 30 days p.i., 10 (8.6%) isolates did not cause any clinical signs in the infected mice, and eight isolates (6.7%) exhibited intermediate

**Table 1. Isolation of *Toxoplasma gondii* from free-range chickens in the Amazon region.**

| Country | State | No. of municipalities | No. of chickens | | No. bioassayed | |
|---|---|---|---|---|---|---|
| | | | Total | Seropositive[a] (%) | Total | No. of isolates (%) |
| Brazil | | | | | | |
| | Acre | 9 | 47 | 41 (87.2) | 32 | 18 (56.2) |
| | Amazonas | 6 | 99 | 76 (76.8) | 62 | 29 (46.8) |
| | Amapá | 6 | 60 | 54 (90.0) | 46 | 25 (54.3) |
| | Pará | 6 | 126 | 55 (43.6) | 40 | 23 (57.5) |
| | Roraima | 2 | 27 | 18 (66.6) | 14 | 8 (57.1) |
| Bolivia | – | 3 | 10 | 8 (80.0) | 6 | 2 (33.3) |
| Guyana | – | 1 | 13 | 8 (61.5) | 7 | 4 (57.1) |
| Peru | – | 2 | 9 | 6 (66.6) | 6 | 4 (66.6) |
| Venezuela | – | 1 | 10 | 7 (70.0) | 7 | 3 (42.8) |
| **Total** | | **36** | **401** | **273 (68.1)** | **220** | **116 (52.7)** |

[a]Modified Agglutination Test (MAT titer ≥ 5 is considered seropositive).

**Table 2. Success rates of *Toxoplasma gondii* isolation versus MAT titers in free-range chickens from this study.**

| Antibody titer (MAT[a]) | No. of chickens[b] | Bioassays | | |
|---|---|---|---|---|
| | | No. of chickens assayed | No. of isolates obtained | % of success |
| 5 | 59 | 12 | 0 | 0.0 |
| 10 | 28 | 25 | 0 | 0.0 |
| 20 | 18 | 17 | 3 | 17.6 |
| 40 | 19 | 19 | 7 | 36.8 |
| 80 | 26 | 26 | 16 | 61.5 |
| 160 | 45 | 45 | 32 | 71.1 |
| 320 | 25 | 24 | 20 | 83.3 |
| 640 | 24 | 24 | 18 | 75.0 |
| 1,280 | 17 | 16 | 13 | 81.2 |
| 2,560 | 6 | 6 | 3 | 50.0 |
| 5,120 | 4 | 4 | 2 | 50.0 |
| 10,240 | 2 | 2 | 2 | 100.0 |
| **Total** | **273** | **220** | **116** | **52.7** |

[a]Modified Agglutination Test (MAT titer ≥ 5 is considered seropositive).

[b]Only 20.3% of chickens were bioassayed for MAT titers at 5.

Linear regression analysis of log (% of success) versus MAT titers from 5 to 1,280 revealed Y = 39.6X−28, whereas Y is the success rate in percentage, X is the log(titers). The adjusted R square = 0.89, with the P value <0.0001. MAT titers from 2,560 to 10,240 are excluded from analysis due to small sample sizes.

behavior; according to these parameters, these isolates were biologically estimated to be virulent, non-virulent, and intermediately virulent, respectively (Table 3).

## Multilocus nested-PCR-RFLP genotyping

A total of 42 non-archetypal PCR-RFLP genotypes were identified among the 116 *T. gondii* isolates obtained in the present study (Tables 3, 4 and Fig 2), and of these, 34 in Brazilian states and eight in the other South American countries that were studied. Of the 42 genotypes, 22 (18 in Brazil and four distributed in Guyana, Venezuela, and Bolivia) were previously described and represent 83 isolates in Brazil and seven in three countries (Guyana, three isolates; Venezuela, three; and Bolivia, one). Twenty genotypes were new and designated as PCR-RFLP genotypes #324–#343. Sixteen of the 20 new genotypes have only one isolate. Sixteen new genotypes were distributed across the five Brazilian states and corresponded to 20 isolates including five new genotypes from Amazonas, four from Acre, three from Amapá, three from Pará, and one from Roraima. The other four new genotypes corresponding to six isolates including two new genotypes from Peru, one from Bolivia, and one from Guyana.

Distribution of the 42 PCR-RFLP genotypes identified in the Brazilian states and other South American countries is summarized in Fig 3. Population comparison for the four Brazilian states with most isolates (Acre, Amapá, Amazonas, Pará) showed that *T. gondii* genotype composition from Acre is very different from the populations from Amapá, Amazonas and Pará states. The most prevalent genotype was #07 (40 isolates) that was distributed in four of the five Brazilian states studied in the Amazon region (Figs 3 and 4).

Four previously elsewhere reported genotypes were identified for the first time in Brazil, including #97, #98, #194, and #219, and seven previously reported genotypes in Brazil were identified for the first time in the Brazilian Amazon region. These included #11 (Type BrII), #19, #57, #71, #94, #109, and #277.

**Table 3.** *Toxoplasma gondii* isolates from free-range chickens of the Amazon region according to locality, serologic titers, virulence in mice and multilocus PCR-RFLP genotype.

| Isolate ID | State[a]/ Country | Municipality | MAT[b] titre | Sick/Infected[c] | % Sick mice | Day of euthanasia p.i.[e] | Mean[f] | Virulence[g] | PCR-RFLP genotype[h] |
|---|---|---|---|---|---|---|---|---|---|
| TgCkBrAC1 | AC | Rio Branco | 640 | 3/3 | 100 | 10[d], 10[d], 10[d] | 10 | VIR | #335 |
| TgCkBrAC2 | AC | Rio Branco | 5120 | 0/2 | 0 | NA[i] | NA | **NVIR** | #336 |
| TgCkBrAC3 | AC | Rio Branco | 10240 | 0/3 | 0 | NA | NA | **NVIR** | #336 |
| TgCkBrAC4 | AC | Plácido de Castro | 80 | 2/3 | 66 | 15, 15 | 15 | VIR | #19 |
| TgCkBrAC5 | AC | Plácido de Castro | 160 | 0/3 | 0 | NA | NA | **NVIR** | #57 |
| TgCkBrAC6 | AC | Acrelândia | 160 | 3/3 | 100 | 13, 13, 13 | 13 | VIR | #41 |
| TgCkBrAC7 | AC | Acrelândia | 640 | 3/3 | 100 | 17, 18, 22 | 19 | VIR | #19 |
| TgCkBrAC8 | AC | Acrelândia | 640 | 3/3 | 100 | 13, 15, 15 | 14 | VIR | #219 |
| TgCkBrAC9 | AC | Acrelândia | 320 | 3/3 | 100 | 14, 15, 15 | 15 | VIR | #219 |
| TgCkBrAC10 | AC | Sena Madureira | 160 | 3/3 | 100 | 19, 21, 21 | 20 | VIR | #11 (BrII) |
| TgCkBrAC11 | AC | Sena Madureira | 160 | 3/3 | 100 | 12, 16, 22 | 17 | VIR | #337 |
| TgCkBrAC12 | AC | Sena Madureira | 160 | 3/3 | 100 | 11[d], 11, 12 | 11 | VIR | #97 |
| TgCkBrAC13 | AC | Sena Madureira | 160 | 3/3 | 100 | 10[d], 10, 11 | 10 | VIR | #97 |
| TgCkBrAC14 | AC | Epitaciolândia | 1280 | 3/3 | 100 | 13, 15, 16 | 15 | VIR | #194 |
| TgCkBrAC15 | AC | Epitaciolândia | 320 | 3/3 | 100 | 15, 17,19 | 17 | VIR | #194 |
| TgCkBrAC16 | AC | Epitaciolândia | 80 | 3/3 | 100 | 14, 15, 19 | 16 | VIR | #194 |
| TgCkBrAC17 | AC | Epitaciolândia | 1280 | 3/3 | 100 | 18, 24, 32 | 25 | VIR | #338 |
| TgCkBrAC18 | AC | Assis Brasil | 1280 | 0/2 | NA | NA | NA | **NVIR** | #8 (BrIII) |
| TgCkBrAM1 | AM | Parintins | 1280 | 3/3 | 100 | 19, 19, 21 | 20 | VIR | #327 |
| TgCkBrAM2 | AM | Parintins | 1280 | 3/3 | 100 | 15, 17, 17 | 16 | VIR | #70 |
| TgCkBrAM3 | AM | Parintins | 80 | 2/2 | 100 | 26, 27 | 27 | VIR | #70 |
| TgCkBrAM4 | AM | Parintins | 1280 | 2/2 | 100 | 21, 22 | 22 | VIR | #327 |
| TgCkBrAM5 | AM | Parintins | 1280 | 2/2 | 100 | 18, 23 | 21 | VIR | #327 |
| TgCkBrAM6 | AM | Parintins | 640 | 1/1 | 100 | 21 | 21 | VIR | #328 |
| TgCkBrAM7 | AM | Parintins | 80 | 1/1 | 100 | 25 | 25 | VIR | #7 |
| TgCkBrAM8 | AM | Parintins | 20 | 3/3 | 100 | 15, 17, 19 | 17 | VIR | #7 |
| TgCkBrAM9 | AM | Tabatinga | 640 | 3/3 | 100 | 15, 16, 16 | 16 | VIR | #14 |
| TgCkBrAM10 | AM | Tabatinga | 320 | 2/2 | 100 | 10[d], 11 | 10 | VIR | #329 |
| TgCkBrAM11 | AM | Tabatinga | 160 | 3/3 | 100 | 17, 19, 21 | 19 | VIR | #14 |
| TgCkBrAM12 | AM | Tabatinga | 640 | 3/3 | 100 | 21, 22, 23 | 22 | VIR | #14 |
| TgCkBrAM13 | AM | Tabatinga | 320 | 3/3 | 100 | 19, 19, 22 | 20 | VIR | #14 |
| TgCkBrAM14 | AM | Tabatinga | 320 | 3/3 | 100 | 14, 16, 18 | 16 | VIR | #14 |
| TgCkBrAM15 | AM | Tabatinga | 5120 | 3/3 | 100 | 20, 22, 23 | 22 | VIR | #14 |
| TgCkBrAM16 | AM | Tabatinga | 1280 | 3/3 | 100 | 12, 12, 16 | 13 | VIR | #330 |
| TgCkBrAM17 | AM | Manacapuru | 40 | 3/3 | 100 | 13, 17, 17 | 16 | VIR | #7 |
| TgCkBrAM18 | AM | Manacapuru | 640 | 3/3 | 100 | 13, 14, 16 | 14 | VIR | #7 |
| TgCkBrAM19 | AM | Manacapuru | 640 | 3/3 | 100 | 20, 22, 25 | 22 | VIR | #7 |
| TgCkBrAM20 | AM | Manacapuru | 640 | 3/3 | 100 | 16, 20, 23 | 20 | VIR | #7 |
| TgCkBrAM21 | AM | Manacapuru | 640 | 3/3 | 100 | 19, 21, 21 | 20 | VIR | #7 |
| TgCkBrAM22 | AM | Manacapuru | 320 | 3/3 | 100 | 20, 21, 25 | 22 | VIR | #7 |
| TgCkBrAM23 | AM | Manacapuru | 640 | 3/3 | 100 | 11, 23, 27 | 20 | VIR | #7 |
| TgCkBrAM24 | AM | Iranduba | 160 | 3/3 | 100 | 25, 25, 28 | 26 | VIR | #7 |
| TgCkBrAM25 | AM | Iranduba | 160 | 1/1 | 100 | 22 | 22 | VIR | #7 |
| TgCkBrAM26 | AM | Iranduba | 320 | 3/3 | 100 | 26, 27, 28 | 27 | VIR | #7 |
| TgCkBrAM27 | AM | Iranduba | 320 | 3/3 | 100 | 19, 27, 30 | 25 | VIR | #7 |

*(Continued)*

**Table 3.** (Continued)

| Isolate ID | State[a]/Country | Municipality | MAT[b] titre | Sick/Infected[c] | % Sick mice | Day of euthanasia p.i.[e] | Mean[f] | Virulence[g] | PCR-RFLP genotype[h] |
|---|---|---|---|---|---|---|---|---|---|
| TgCkBrAM28 | AM | Iranduba | 320 | 0/3 | 0 | NA | NA | **NVIR** | #8 (BrIII) |
| TgCkBrAM29 | AM | Rio Preto da Eva | 160 | 2/2 | 100 | 9[d], 9[d] | 9 | VIR | #331 |
| TgCkBrAP1 | AP | Mazagão | 2560 | 3/3 | 100 | 11, 12, 13 | 12 | VIR | #98 |
| TgCkBrAP2 | AP | Mazagão | 160 | 3/3 | 100 | 17, 22, 27 | 22 | VIR | #7 |
| TgCkBrAP3 | AP | Mazagão | 10240 | 3/3 | 100 | 18, 18, 19 | 18 | VIR | #7 |
| TgCkBrAP4 | AP | Mazagão | 640 | 1/1 | 100 | 22 | 22 | VIR | #7 |
| TgCkBrAP5 | AP | Mazagão | 80 | 0/1 | 0 | NA | NA | **NVIR** | #7 |
| TgCkBrAP6 | AP | Mazagão | 160 | 3/3 | 100 | 20, 28, 36 | 28 | VIR | #7 |
| TgCkBrAP7 | AP | Macapá | 80 | 0/3 | 0 | NA | NA | **NVIR** | #28 |
| TgCkBrAP8 | AP | Macapá | 640 | 2/3 | 67 | 24, 30 | 27 | VIR | #28 |
| TgCkBrAP9 | AP | Macapá | 1280 | 3/3 | 100 | 16, 16, 21 | 18 | VIR | #70 |
| TgCkBrAP10 | AP | Itaubal | 1280 | 3/3 | 100 | 11, 12, 13 | 12 | VIR | #332 |
| TgCkBrAP11 | AP | Itaubal | 320 | 0/1 | 0 | NA | NA | **NVIR** | #28 |
| TgCkBrAP12 | AP | Tartarugalzinho | 160 | 3/3 | 100 | 16, 16, 17 | 16 | VIR | #71 |
| TgCkBrAP13 | AP | Tartarugalzinho | 320 | 2/3 | 67 | 20, 27 | 23 | VIR | #28 |
| TgCkBrAP14 | AP | Tartarugalzinho | 80 | 3/3 | 100 | 16, 16, 21 | 18 | VIR | #333 |
| TgCkBrAP15 | AP | Tartarugalzinho | 2560 | 3/3 | 100 | 21, 26, 30 | 26 | VIR | #70 |
| TgCkBrAP16 | AP | Tartarugalzinho | 320 | 3/3 | 100 | 19, 20, 37 | 25 | VIR | #97 |
| TgCkBrAP17 | AP | Ferreira Gomes | 80 | 2/2 | 100 | 20, 28 | 24 | VIR | #71 |
| TgCkBrAP18 | AP | Ferreira Gomes | 2560 | 3/3 | 100 | 21, 23, 29 | 24 | VIR | #71 |
| TgCkBrAP19 | AP | Ferreira Gomes | 160 | 3/3 | 100 | 20, 21, 24 | 22 | VIR | #7 |
| TgCkBrAP20 | AP | Ferreira Gomes | 80 | 3/3 | 100 | 17, 23, 28 | 23 | VIR | #28 |
| TgCkBrAP21 | AP | Ferreira Gomes | 40 | 3/3 | 100 | 20, 22, 34 | 25 | VIR | #7 |
| TgCkBrAP22 | AP | Ferreira Gomes | 1280 | 3/3 | 100 | 17, 18, 19 | 18 | VIR | #7 |
| TgCkBrAP23 | AP | Porto Grande | 40 | 3/3 | 100 | 12, 14, 15 | 14 | VIR | #109 |
| TgCkBrAP24 | AP | Porto Grande | 640 | 1/2 | 50 | 20 | 20 | INT | #109 |
| TgCkBrAP25 | AP | Macapá | 160 | 3/3 | 100 | 15, 24, 26 | 22 | VIR | #28 |
| TgCkBrPA1 | PA | Monte Alegre | 20 | 1/3 | 33 | 20 | 20 | INT | #7 |
| TgCkBrPA2 | PA | Monte Alegre | 160 | 2/2 | 100 | 15, 15 | 15 | VIR | #70 |
| TgCkBrPA3 | PA | Monte Alegre | 320 | 4/4 | 100 | 14, 19, 19, 22 | 19 | VIR | #7 |
| TgCkBrPA4 | PA | Monte Alegre | 160 | 2/2 | 100 | 22, 25 | 23 | VIR | #7 |
| TgCkBrPA5 | PA | Monte Alegre | 40 | 4/4 | 100 | 14, 18, 19, 24 | 19 | VIR | #70 |
| TgCkBrPA6 | PA | Monte Alegre | 320 | 4/4 | 100 | 15, 17, 18, 24 | 19 | VIR | #7 |
| TgCkBrPA7 | PA | Monte Alegre | 160 | 4/4 | 100 | 19, 19, 20, 22 | 20 | VIR | #7 |
| TgCkBrPA8 | PA | Monte Alegre | 80 | 3/3 | 100 | 27, 34, 39 | 33 | INT | #7 |
| TgCkBrPA9 | PA | Monte Alegre | 80 | 5/5 | 100 | 17, 17, 18, 18, 22 | 18 | VIR | #324 |
| TgCkBrPA10 | PA | Alenquer | 160 | 3/3 | 100 | 24, 27, 27 | 26 | VIR | #7 |
| TgCkBrPA11 | PA | Alenquer | 1280 | 3/3 | 100 | 22, 24, 24 | 23 | VIR | #7 |
| TgCkBrPA12 | PA | Alenquer | 320 | 1/2 | 50 | 24 | 24 | INT | #7 |
| TgCkBrPA13 | PA | Alenquer | 80 | 3/3 | 100 | 12, 15, 16 | 14 | VIR | #7 |
| TgCkBrPA14 | PA | Alenquer | 80 | 2/2 | 100 | 20, 21 | 20 | VIR | #7 |
| TgCkBrPA15 | PA | Alenquer | 320 | 2/2 | 100 | 32, 32 | 32 | INT | #7 |
| TgCkBrPA16 | PA | Alenquer | 80 | 3/3 | 100 | 21, 21, 24 | 22 | VIR | #7 |
| TgCkBrPA17 | PA | Soure | 160 | 3/3 | 100 | 16, 16, 16 | 16 | VIR | #258 |
| TgCkBrPA18 | PA | Soure | 160 | 3/3 | 100 | 13, 13, 13 | 13 | VIR | #325 |
| TgCkBrPA19 | PA | Soure | 1280 | 3/3 | 100 | 24, 27, 31 | 27 | VIR | #28 |

*(Continued)*

**Table 3.** (Continued)

| Isolate ID | State[a]/Country | Municipality | MAT[b] titre | Sick/Infected[c] | % Sick mice | Day of euthanasia p.i.[e] | Mean[f] | Virulence[g] | PCR-RFLP genotype[h] |
|---|---|---|---|---|---|---|---|---|---|
| TgCkBrPA20 | PA | Soure | 160 | 0/3 | NA | NA | NA | **NVIR** | #28 |
| TgCkBrPA21 | PA | Soure | 640 | 3/3 | 100 | 20, 21, 27 | 23 | VIR | #326 |
| TgCkBrPA22 | PA | Soure | 160 | 3/3 | 100 | 15, 18, 24 | 19 | VIR | #326 |
| TgCkBrPA23 | PA | Soure | 320 | 3/3 | 100 | 11, 12, 12 | 12 | VIR | #94 |
| TgCkBrRR1 | RR | Boa vista | 160 | 3/3 | 100 | 15, 23, 23 | 20 | VIR | #7 |
| TgCkBrRR2 | RR | Boa vista | 160 | 3/3 | 100 | 20, 20, 23 | 21 | VIR | #7 |
| TgCkBrRR3 | RR | Boa vista | 160 | 3/3 | 100 | 17, 19, 20 | 19 | VIR | #7 |
| TgCkBrRR4 | RR | Boa vista | 40 | 3/3 | 100 | 16, 19, 23 | 19 | VIR | #7 |
| TgCkBrRR5 | RR | Boa vista | 640 | 2/2 | 100 | 21, 23 | 22 | VIR | #7 |
| TgCkBrRR6 | RR | Boa vista | 80 | 1/2 | 50 | 26 | 26 | INT | #7 |
| TgCkBrRR7 | RR | Boa vista | 40 | 2/3 | 67 | 13, 16 | 15 | VIR | #339 |
| TgCkBrRR8 | RR | Bonfim | 320 | 3/3 | 100 | 14, 16, 17 | 16 | VIR | #277 |
| TgCkBo1 | Bolivia | Villa Busch | 160 | 1/1 | 100 | 25 | 25 | VIR | #95 |
| TgCkBo2 | Bolivia | Villa Busch | 160 | 1/1 | 100 | 25 | 25 | VIR | #340 |
| TgCkGy37 | Guyana | Lethem | 20 | 1/3 | 33 | 30 | 30 | INT | #12 |
| TgCkGy38 | Guyana | Lethem | 160 | 3/3 | 100 | 15, 16, 16 | 16 | VIR | #343 |
| TgCkGy39 | Guyana | Lethem | 40 | 1/1 | 100 | 44 | 44 | INT | #12 |
| TgCkGy40 | Guyana | Lethem | 640 | 0/3 | 0 | NA | NA | **NVIR** | #12 |
| TgCkPe11 | Peru | Iñapari | 640 | 3/3 | 100 | 10, 22, 30 | 21 | VIR | #341 |
| TgCkPe12 | Peru | Iñapari | 80 | 2/2 | 100 | 7[d], 23 | 15 | VIR | #341 |
| TgCkPe13 | Peru | Iñapari | 320 | 3/3 | 100 | 14, 17, 19 | 17 | VIR | #341 |
| TgCkPe14 | Peru | Noaya | 160 | 2/2 | 100 | 16, 16 | 16 | VIR | #342 |
| TgCkVe14 | Venezuela | Santa Elena de Uairén | 320 | 3/3 | 100 | 13, 14, 14 | 14 | VIR | #48 |
| TgCkVe15 | Venezuela | Santa Elena de Uairén | 160 | 3/3 | 100 | 13, 15, 15 | 14 | VIR | #48 |
| TgCkVe16 | Venezuela | Santa Elena de Uairén | 160 | 3/3 | 100 | 11, 12, 13 | 12 | VIR | #123 |

[a]Brazilian states: AC = Acre, AM = Amazonas, AP = Amapá, PA = Pará, RR = Roraima.

[b]MAT = Modified Agglutination Test (titer ≥ 5 is considered seropositive).

[c]Mice were considered infected with the parasite when tachyzoites or tissue cysts were present in their tissues using direct microscopic examination.

[d]Mice died of toxoplasmosis.

[e]p.i. = post-inoculation.

[f]Mean day of euthanasia p.i.

[g]VIR = virulent: acute sickness observed in more than 50% of the infected mice within 30 days p.i.; **NVIR** = non-virulent: no clinical signs observed in the infected mice; INT = intermediate virulence: intermediate behavior.

[h]#324 to #343 represent new genotypes.

[i]NA = not applicable.

To reveal the genetic relationship of *T. gondii* isolates obtained in this study to previously reported isolates, we conducted phylogenetic network analysis of 144 PCR-RFLP genotypes from 504 isolates. These genotypes included 81 genotypes (from 218 isolates) previously reported in the Southeastern region of Brazil [6,9,27,36–68], 52 genotypes (from 116 isolates from this study and 19 previously reported [6,21,44,61,69,70]) in the Northern region of Brazil; nine genotypes occurring in both the Southeastern and Northern regions of Brazil (123 isolates from Southeastern and 28 from Northern regions), and archetypal Type I and Type II variant genotypes (S3 Table).

Table 4. Genotyping of *Toxoplasma gondii* isolates from free-range chickens from Amazon region, using Multilocus nested-PCR-RFLP.

| Isolate ID | Country/State[a] | Munic.[b] | SAG1 | SAG2 (5′3′) | SAG2 (alt.) | SAG3 | BTUB | GRA6 | c22-8 | c29-2 | L358 | PK1 | Apico | CS3[c] | Genotype ToxoDB-RFLP |
|---|---|---|---|---|---|---|---|---|---|---|---|---|---|---|---|
| TgCkBrPA1,3,4,6,7,8 | Br/PA | Monte Alegre | I | III | III | III | III | III | III | III | III | III | I | I | #07 |
| 10,11,12,13,14,15,16 | | Alenquer | | | | | | | | | | | | | (n = 40) |
| TgCkBrAM7,8 | Br/AM | Parintins | | | | | | | | | | | | | |
| 17,18,19,20,21,22,23 | | Manacapuru | | | | | | | | | | | | | |
| 24,25,26,27 | | Iranduba | | | | | | | | | | | | | |
| TgCkBrAP2,3,4,5,6 | Br/AP | Mazagão | | | | | | | | | | | | | |
| 19,21,22 | | Ferreira Gomes | | | | | | | | | | | | | |
| TgCkBrRR1,2,3,4,5,6 | Br/RR | Boa Vista | | | | | | | | | | | | | |
| TgCkBrAM28 | Br/AM | Iranduba | I | III | III | III | III | III | II | III | III | III | III | III | #08 |
| TgCkBrAC18 | Br/AC | Assis Brasil | | | | | | | | | | | | | (BrIII) (n = 2) |
| TgCkBrAC10 | Br/AC | Sena Madureira | I | I | II | III | III | III | I | III | I | II | III | I | #11 |
| | | | | | | | | | | | | | | | (BrII) (n = 1) |
| TgCkBrAM9,11,12,13,14,15 | Br/AM | Tabatinga | I | III | III | III | III | III | III | I | III | III | III | II | #14 |
| | | | | | | | | | | | | | | | (n = 6) |
| TgCkBrAC4 | Br/AC | Plácido de Castro | I | III | III | III | III | III | I | I | I | u-1 | I | II | #19 |
| 7 | | Acrelândia | | | | | | | | | | | | | (n = 2) |
| TgCkBrPA19,20 | Br/PA | Soure | I | I | I | I | I | I | II | I | III | I | III | III | #28 |
| TgCkBrAP7,8,25 | Br/AP | Macapá | | | | | | | | | | | | | (n = 7) |
| 11 | | Itaubal | | | | | | | | | | | | | |
| 13 | | Tartarugalzinho | | | | | | | | | | | | | |
| TgCkBrAC6 | Br/AC | Acrelândia | I | I | I | III | I | II | I | I | I | I | I | I | #41 |
| | | | | | | | | | | | | | | | (n = 1) |
| TgCkBrAC5 | Br/AC | Plácido de Castro | I | I | I | III | I | II | u-1 | I | III | II | III | III | #57 |
| | | | | | | | | | | | | | | | (n = 1) |
| TgCkBrPA2,5 | Br/PA | Monte Alegre | I | III | III | III | III | II | u-1 | I | I | I | III | II | #70 |
| TgCkBrAM2,3 | Br/AM | Parintins | | | | | | | | | | | | | (n = 6) |
| TgCkBrAP9 | Br/AP | Macapá | | | | | | | | | | | | | |
| 15 | | Tartarugalzinho | | | | | | | | | | | | | |
| TgCkBrAP12 | Br/AP | Tartarugalzinho | I | III | I | III | III | III | II | I | III | III | I | I | #71 |
| 17,18 | | Ferreira Gomes | | | | | | | | | | | | | (n = 3) |
| TgCkBrPA23 | Br/PA | Soure | I | I | II | I | III | III | I | I | I | II | I | III | #94 |
| | | | | | | | | | | | | | | | (n = 1) |

*(Continued)*

Table 4. (Continued)

| Isolate ID | [a]Country/State | Munic.[b] | PCR-RFLP Markers | | | | | | | | | | | | Genotype ToxoDB- |
| | | | SAG1 | 5′3′ SAG2 | alt. SAG2 | SAG3 | BTUB | GRA6 | c22-8 | c29-2 | L358 | PK1 | Apico | CS3[c] | RFLP |
|---|---|---|---|---|---|---|---|---|---|---|---|---|---|---|---|
| **TgCkBrAP16** | Br/AP | Tartarugalzinho | I | I | II | I | III | III | II | III | I | III | I | **I** | **#97** |
| **TgCkBrAC12,13** | Br/AC | Sena Madureira | I | I | II | I | III | III | II | III | I | III | I | **u-3** | **#97[d]** |
| | | | | | | | | | | | | | | | (n = 3) |
| **TgCkBrAP1** | Br/AP | Mazagão | I | I | II | I | III | III | III | III | I | III | III | II | **#98** |
| | | | | | | | | | | | | | | | (n = 1) |
| **TgCkBrAP23,24** | Br/AP | Porto Grande | I | I | II | I | III | III | III | I | III | III | III | II | **#109** |
| | | | | | | | | | | | | | | | (n = 2) |
| **TgCkBrAC14,15,16** | Br/AC | Epitaciolândia | I | I | II | I | III | III | II | I | II | III | III | I | **#194** |
| | | | | | | | | | | | | | | | (n = 3) |
| **TgCkBrAC8,9** | Br/AC | Acrelândia | II/III | III | III | III | III | III | III | III | III | III | I | II | **#219** |
| | | | | | | | | | | | | | | | (n = 2) |
| **TgCkBrPA17** | Br/PA | Soure | I | I | II | I | III | III | II | III | I | III | I | I | **#258** |
| | | | | | | | | | | | | | | | (n = 1) |
| **TgCkBrRR8** | Br/RR | Bonfim | I | III | III | I | III | III | II | III | III | I | III | III | **#277** |
| | | | | | | | | | | | | | | | (n = 1) |
| **TgCkGy37,39,40** | Gy | Lethem | I | III | III | I | III | III | II | III | III | I | III | III | **#12** |
| | | | | | | | | | | | | | | | (n = 3) |
| **TgCkVe14,15** | Ve | Sta. Elena de Uairén | I | III | III | III | III | III | III | III | III | III | III | I | **#48** |
| | | | | | | | | | | | | | | | (n = 2) |
| **TgCkBo1** | Bo | Villa Busch | I | I | II | I | III | III | II | I | I | III | I | II | **#95** |
| | | | | | | | | | | | | | | | (n = 1) |
| **TgCkVe16** | Ve | Sta. Elena de Uairén | I | III | III | III | III | III | I | III | III | III | III | u-1 | **#123** |
| | | | | | | | | | | | | | | | (n = 1) |
| **TgCkBrPA9** | Br/PA | Monte Alegre | I | I | II | I | III | III | u-1 | III | I | III | I | II | **#324, new** |
| **TgCkBrPA18** | Br/PA | Soure | I | I | I | I | I | II | II | I | I | I | III | II | **#325, new** |
| **TgCkBrPA21,22** | Br/PA | Soure | I | I | I | I | I | I | II | I | I | I | III | I | **#326, new** |
| **TgCkBrAM1,4,5** | Br/AM | Parintins | I | III | III | III | III | III | III | III | III | I | III | II | **#327, new** |
| **TgCkBrAM6** | Br/AM | Parintins | I | III | III | I | III | III | I | III | III | III | I | u-1 | **#328, new** |
| **TgCkBrAM10** | Br/AM | Tabatinga | I | I | II | I | III | III | I | III | I | III | I | I | **#329, new** |
| **TgCkBrAM16** | Br/AM | Tabatinga | I | I | I | I | III | I | I | I | III | I | III | I | **#330, new** |
| **TgCkBrAM29** | Br/AM | R. Preto Eva | I | II | II | I | III | III | II | I | III | u-1 | I | I | **#331, new** |
| **TgCkBrAP10** | Br/AP | Itaubal | I | I | II | I | I | III | II | III | III | III | I | I | **#332, new** |
| **TgCkBrAP14** | Br/AP | Tartarugalzinho | I | I | II | I | I | III | u-1 | III | III | III | I | I | **#333, new** |
| **TgCkBrAP20** | Br/AP | Ferreira Gomes | I | III | III | III | III | II | u-1 | III | III | I | III | II | **#334, new** |

*(Continued)*

Table 4. (Continued)

| Isolate ID | Country/State[a] | Munic.[b] | PCR-RFLP Markers | | | | | | | | | | | | Genotype ToxoDB-RFLP |
|---|---|---|---|---|---|---|---|---|---|---|---|---|---|---|---|
| | | | SAG1 | 5'3' SAG2 | alt. SAG2 | SAG3 | BTUB | GRA6 | c22-8 | c29-2 | L358 | PK1 | Apico | CS3[c] | |
| TgCkBrAC1 | Br/AC | Rio Branco | I | I | II | I | III | III | III | I | II | III | I | u-1 | #335, new |
| TgCkBrAC2,3 | Br/AC | Rio Branco | I | III | III | III | I | III | I | III | III | I | III | III | #336, new |
| TgCkBrAC11 | Br/AC | Sena Madureira | I | I | II | I | III | III | III | I | II | II | III | II | #337, new |
| TgCkBrAC17 | Br/AC | Epitaciolândia | u-1 | I | II | I | III | III | III | I | III | III | I | I | #338, new |
| TgCkBrRR7 | Br/RR | Boa Vista | I | I | II | III | III | III | II | I | II | III | I | I | #339, new |
| TgCkBo2 | Bo | Villa Busch | I | I | I | III | I | I | I | I | I | I | III | I | #340, new |
| TgCkPe11,12,13 | Pe | Iñapari | I | I | II | I | I | III | II | I | II | III | III | II | #341, new |
| TgCkPe14 | Pe | Noaya | I | I | II | I | III | III | u-1 | I | I | u-2 | I | I | #342, new |
| TgCkGy38 | Gy | Lethem | I | I | II | I | III | III | II | III | III | III | I | II | #343, new |

[a]Br = Brazil; Bo = Bolivia; Gy = Guyana; Pe = Peru; Ve = Venezuela; AC = Acre state; AM = Amazonas state; AP = Amapá state; PA = Pará state; RR = Roraima state.

[b]Munic. = Municipality.

[c]CS3 marker was not used to identify a genotype, but variants of the same genotype.

[d]This genotype is a CS3 variant.

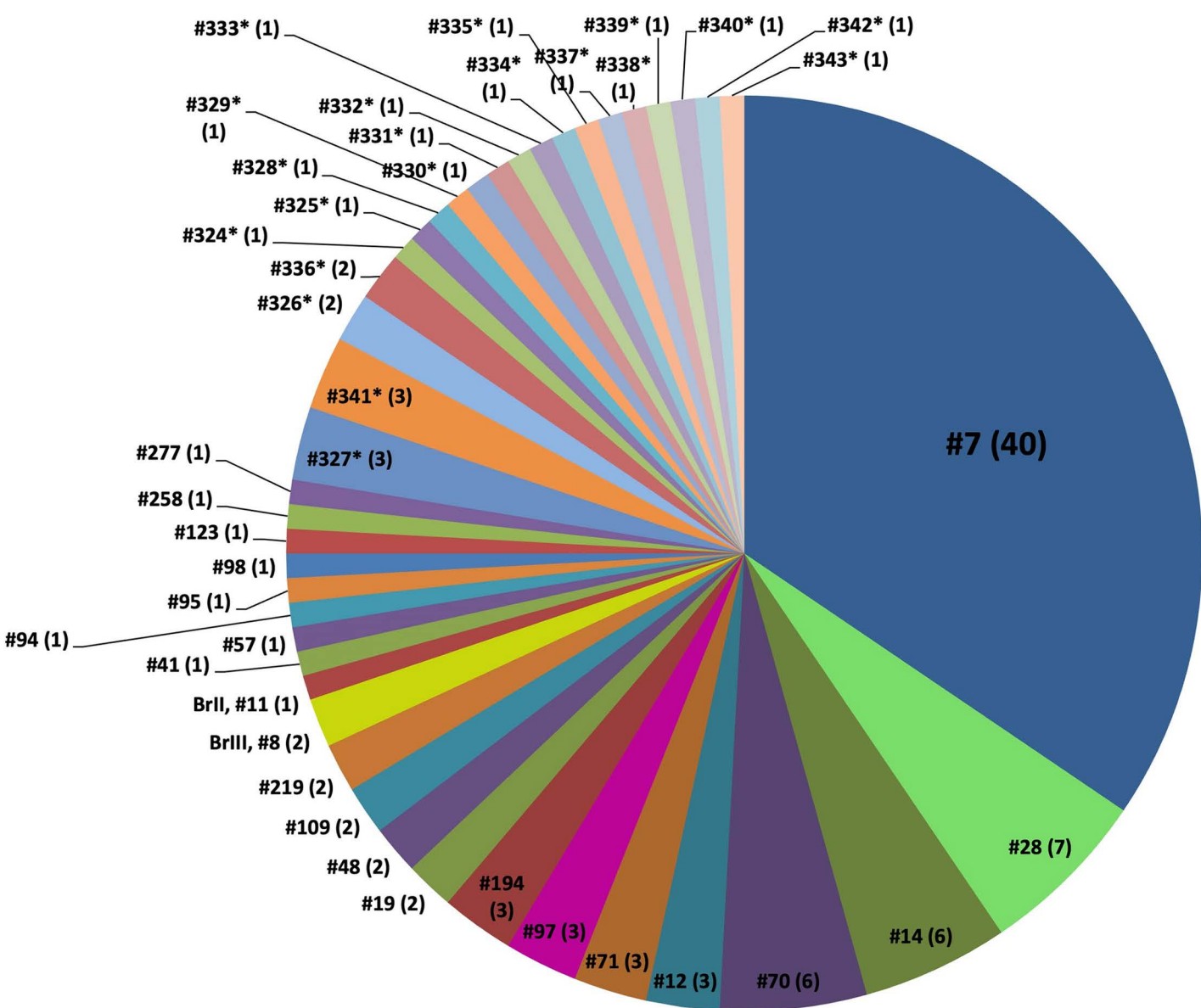

**Fig 2. Frequency of each multilocus PCR-RFLP genotype identified among the *Toxoplasma gondii* isolates from free-range chickens in the Amazon region.** A total of 42 genotypes were identified from 116 *T. gondii* isolates. The number in the parenthesis represents the number of isolates belonging to the genotype. Genotypes #324* to #341* are new genotypes not previously reported.

The results of the phylogenetic network analysis are presented in Fig 5. The 144 genotypes were divided into five groups with two major clusters (Groups 1 and 5) (Fig 5). Group 1 contained 34 genotypes (49 isolates), and Group 5 contained 55 genotypes (240 isolates). Most of the genotypes observed in the Amazon region in this study and from other studies in the Northern region of Brazil [6,21,44,61,69,70], were clustered in Group 1 (29 genotypes, 44 isolates), and most of the genotypes described in the Southeastern region of Brazil were clustered in Group 5 (51 genotypes, 230 isolates). Types BrI and BrII were in Group 5, Types III and BrIII were in Group 3, Type I was in Group 2, and Type II and Type II variant were in Group 1. The most prevalent #7 genotype was observed in Group 3.

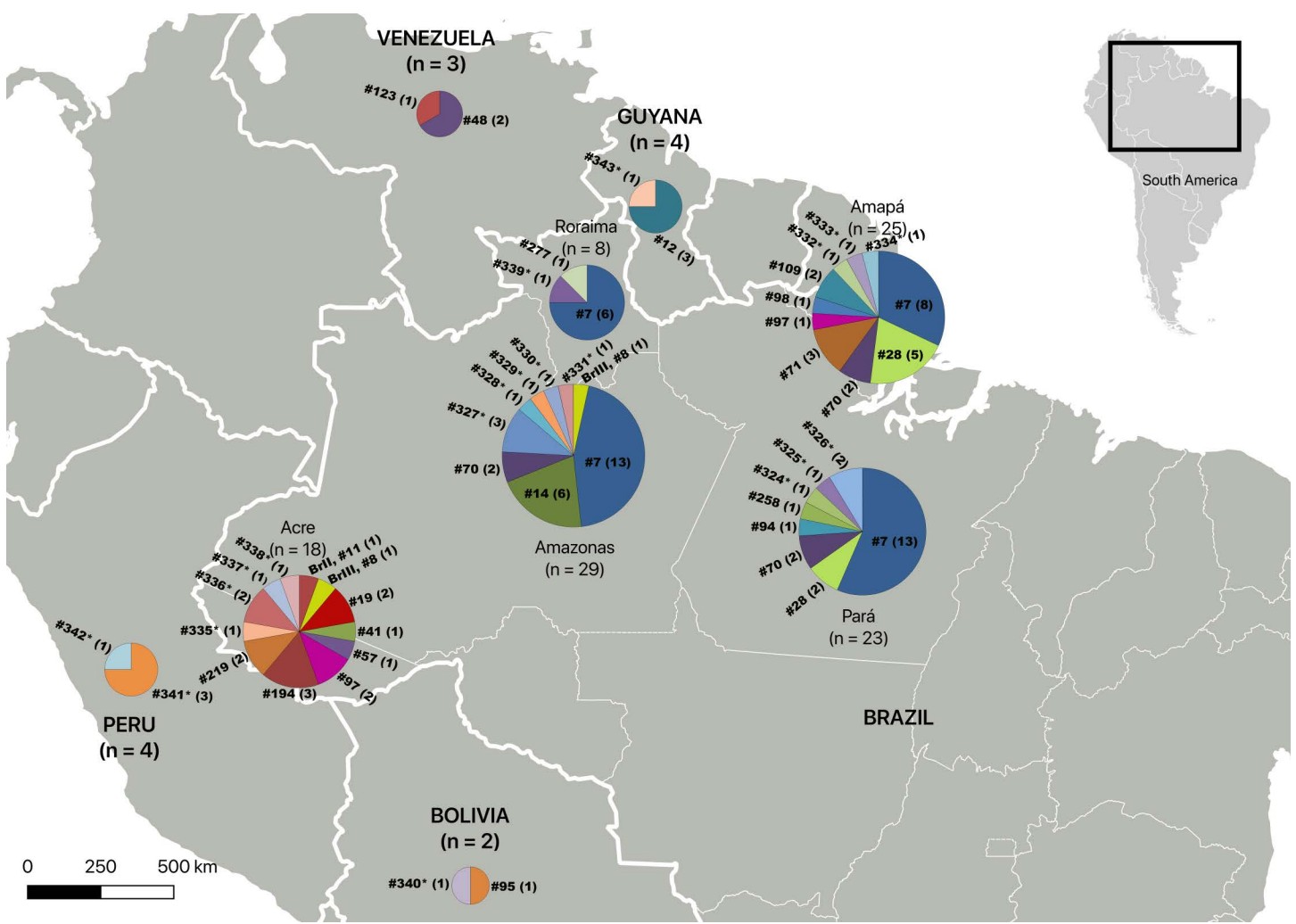

**Fig 3.** *Toxoplasma gondii* **sample sizes and genotype distribution among the five Brazilian states and four other South American countries.** The five Brazilian states include Acre, Amapá, Amazonas, Pará and Roraima, and the four other South American countries include Bolivia, Guyana, Peru, and Venezuela. The number in the parenthesis represents the number of isolates belonging to the genotype. Genotypes #324* to #341* are new genotypes not previously reported. Data were plotted using QGIS software (basemap from https://www.ibge.gov.br/geociencias/organizacao-do-territorio/malhas-territoriais/15774-malhas.html).

Overall, the PCR-RFLP genotypes identified in Northern Brazilian states from this study were different from those identified in the other South American countries previously reported (Fig 5).

Relative to the CS3 alleles, the observed frequencies of virulence in mice (according to the established ethical criteria) were significantly different from those estimated for allele III (p < 0.001), thus indicating that allele III is associated with lower virulence than alleles I and II.

## Microsatellite analysis

Among the 116 isolates, 38 MS types were observed (considering the eight typing markers), but considering the set of 15 markers, there were 83 genotypes. Of these, 26 genotypes were represented by two to four isolates. All isolates possessed non-archetypal genotypes except for TCkVe16 that was MS archetypal Type III. The MS results are listed in Table 5.

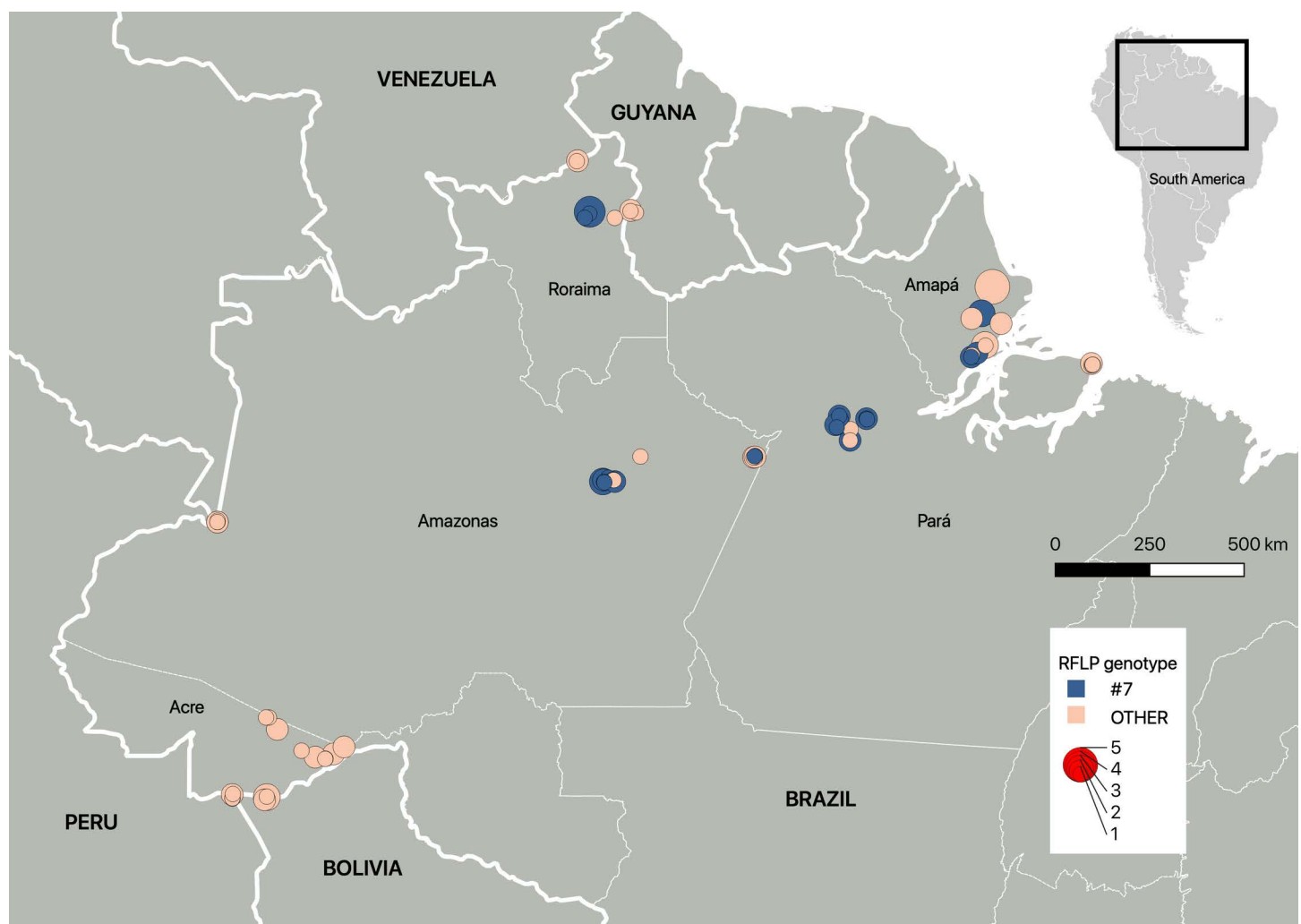

**Fig 4. Map showing the distribution of the most prevalent PCR-RFLP genotype #7 and the other genotypes, according to the respective number of *Toxoplasma gondii* isolates from free-range chickens and GPS coordinates of the collected samples in the Amazon region.** Data were plotted using QGIS software (basemap from https://www.ibge.gov.br/geociencias/organizacao-do-territorio/malhas-territoriais/15774-malhas.html).

Most isolates possessed unique/non-archetypal alleles in one to four typing markers except for TgCkBo1 that had five typing markers (TgMA, B18, B17, M33, and IV.1) with unique alleles. Six isolates possessed only archetypal alleles in the typing markers (TgCk-BrAC10, TgCkBrAM16, TgCkGy37,39,40, and TgCkVe16), 22 isolates possessed one non-archetypal allele, 17 exhibited two non-archetypal alleles, 65 possessed three, and five isolates possessed four non-archetypal alleles. The most prevalent MS type (43 isolates) corresponding particularly to #7 RFLP genotype exhibits three typing markers (TgMA, B18, and B17) with unique alleles.

Unique alleles were observed in all the eight typing markers, that included TUB2 (293), W35 (240, 246), TgMA (203), B18 (156, 162, 164, 168), B17 (334, 340, 344, 348, 362, 364, 350), M33 (167, 171, 173), IV.1 (272, 276) and XI.1 (354). B17 typing marker exhibited greater variability, and only 12 isolates possessed archetypal alleles at the B17 marker (336 or 342).

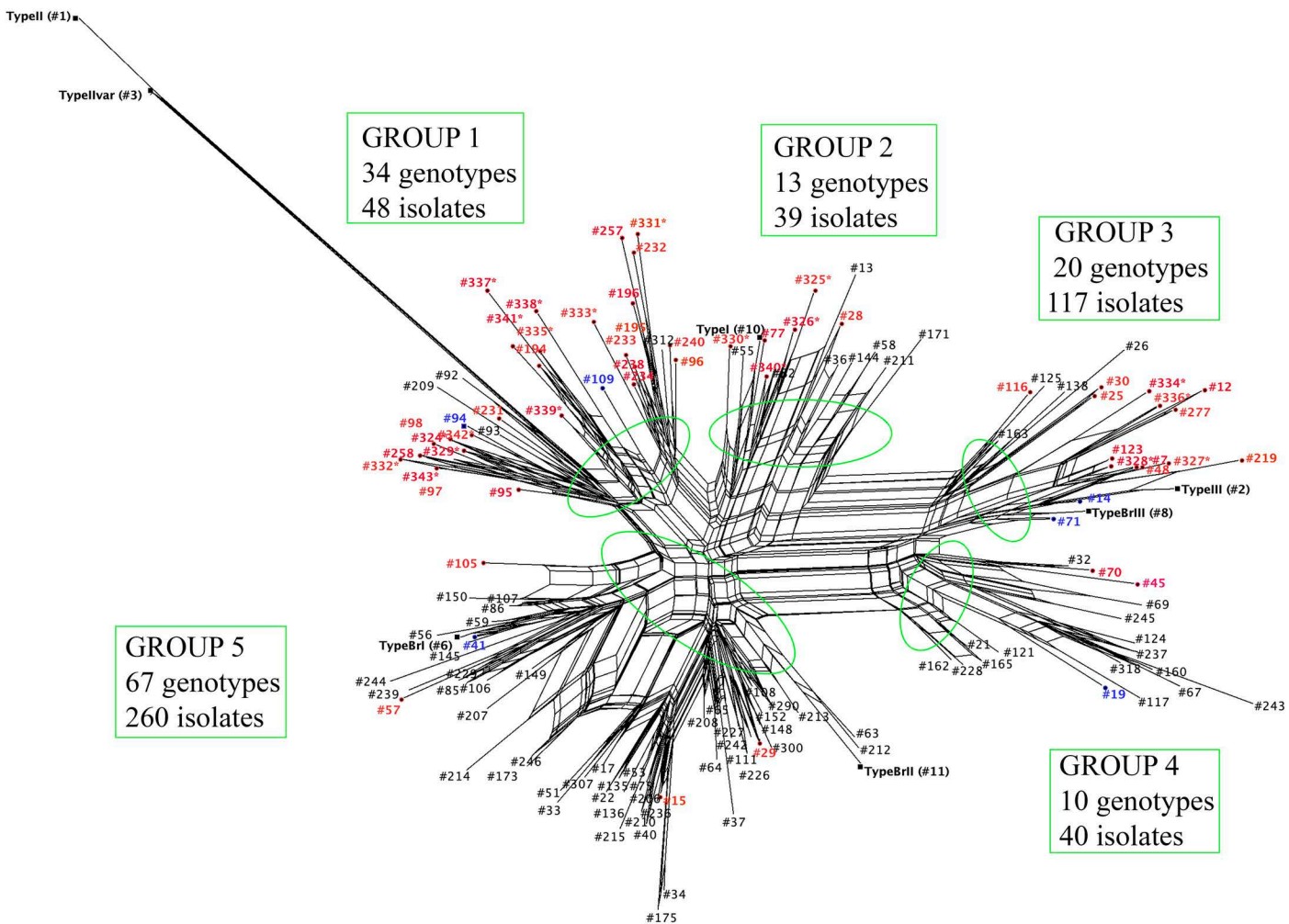

**Fig 5. Phylogenetic network (NeighborNet) of *Toxoplasma gondii* isolates/samples from free-range chickens in the Amazon and the Southeastern regions of Brazil.** A total of 144 PCR-RFLP genotypes from 504 isolates are included in this analysis. Green circles show the major clusters formed. The red labels represent genotypes described in the present study and in previous studies in the North region of Brazil. The blue labels represent genotypes circulating both in the North and the Southeastern regions of Brazil. The black labels represent genotypes previously described in the Southeastern region of Brazil. Black/bold labels are reference genotypes. Genotypes #324* to #341* are new genotypes not previously reported.

## Diversity analysis

The value of *D* for the entire *T. gondii* population used in the present study (116 isolates) was 0.872, that for the Northern region of Brazil (103 isolates from the present study and 47 isolates from previous studies in Pará, Rondônia, and Amazonas states was 0.908, and that for the Southeastern region of Brazil (90 genotypes, 341 isolates) was 0.961.

## Discussion

The population structure of *T. gondii* throughout the world differs, but there are certain genotypes that dominate in the northern hemisphere (North America and Europe) and hundreds circulating in the southern hemisphere [6,10,30]. Information regarding *T. gondii* population structure in Africa and Asia remains limited; however, there are descriptions of major local lineages [3,71].

**Table 5. Genotyping results of 116 *Toxoplasma gondii* isolates from free-range chickens from the Amazon region using microsatellite analysis.**

| Isolate ID | State/ Country[a] | Municipality | RFLP Genotype | MS Type[b] | Microsatellite markers[c,d] | | | | | | | | | | | | | | | N. of types | N. of subtypes |
|---|---|---|---|---|---|---|---|---|---|---|---|---|---|---|---|---|---|---|---|---|---|
| | | | | | TUB2 | W35 | TgM-A | B18 | B17 | M33 | IV.1 | XI.1 | M48 | M102 | N60 | N82 | AA | N61 | N83 | | |
| ENT[e] | – | – | I | 1 | 291 | 248 | 209 | 160 | 342 | 169 | 274 | 358 | 209 | 166 | 145 | 119 | 267 | 87 | 306 | 1 | |
| ME49[e] | – | – | II | II | 289 | 242 | 207 | 158 | 336 | 169 | 274 | 356 | 215 | 174 | 142 | 111 | 265 | 91 | 310 | | |
| NED[e] | – | – | III | III | 289 | 242 | 205 | 160 | 336 | 165 | 278 | 356 | 209 | 190 | 147 | 111 | 267 | 91 | 312 | | |
| TgCkBrPA1 | PA | Monte Alegre | #7 | N. arch. | 289 | 242 | 203 | 156 | 340 | 165 | 278 | 356 | 245 | 170 | 142 | 111 | 271 | 95 | 312 | 1 | 1 |
| TgCkBrPA3 | PA | Monte Alegre | #7 | N. arch. | 289 | 242 | 203 | 156 | 340 | 165 | 278 | 356 | 229 | 170 | 142 | 111 | 271 | 95 | 312 | | 2 |
| TgCkBrPA4 | PA | Monte Alegre | #7 | N. arch. | 289 | 242 | 203 | 156 | 340 | 165 | 278 | 356 | 241 | 170 | 142 | 111 | 271 | 95 | 312 | | 3 |
| TgCkBrPA6 | PA | Monte Alegre | #7 | N. arch. | 289 | 242 | 203 | 156 | 340 | 165 | 278 | 356 | 245 | 170 | 142 | 111 | 269 | 99 | 312 | | 4 |
| TgCkBrPA7 | PA | Monte Alegre | #7 | N. arch. | 289 | 242 | 203 | 156 | 340 | 165 | 278 | 356 | 245 | 170 | 142 | 111 | 269 | 99 | 312 | | 4 |
| TgCkBrPA14 | PA | Alenquer | #7 | N. arch. | 289 | 242 | 203 | 156 | 340 | 165 | 278 | 356 | 245 | 170 | 142 | 111 | 269 | 99 | 312 | | 4 |
| TgCkBrPA8 | PA | Monte Alegre | #7 | N. arch. | 289 | 242 | 203 | 156 | 340 | 165 | 278 | 356 | 247 | 170 | 142 | 111 | 269 | 99 | 312 | | 5 |
| TgCkBrPA10 | PA | Alenquer | #7 | N. arch. | 289 | 242 | 203 | 156 | 340 | 165 | 278 | 356 | 237 | 170 | 142 | 111 | 279 | 95 | 312 | | 6 |
| TgCkBrPA11 | PA | Alenquer | #7 | N. arch. | 289 | 242 | 203 | 156 | 340 | 165 | 278 | 356 | 237 | 170 | 142 | 111 | 279 | 95 | 312 | | 6 |
| TgCkBrPA12 | PA | Alenquer | #7 | N. arch. | 289 | 242 | 203 | 156 | 340 | 165 | 278 | 356 | 241 | 170 | 142 | 111 | 277 | 95 | 312 | | 7 |
| TgCkBrPA13 | PA | Alenquer | #7 | N. arch. | 289 | 242 | 203 | 156 | 340 | 165 | 278 | 356 | 227 | 170 | 142 | 111 | 273 | 99 | 312 | | 8 |
| TgCkBrPA15 | PA | Alenquer | #7 | N. arch. | 289 | 242 | 203 | 156 | 340 | 165 | 278 | 356 | 231 | 170 | 142 | 111 | 271 | 97 | 312 | | 9 |
| TgCkBrPA16 | PA | Alenquer | #7 | N. arch. | 289 | 242 | 203 | 156 | 340 | 165 | 278 | 356 | 237 | 170 | 142 | 111 | 271 | 97 | 312 | | 10 |
| TgCkBrAM7 | AM | Parintins | #7 | N. arch. | 289 | 242 | 203 | 156 | 340 | 165 | 278 | 356 | 233 | 170 | 142 | 111 | 271 | 93 | 312 | | 11 |
| TgCkBrAM8 | AM | Parintins | #7 | N. arch. | 289 | 242 | 203 | 156 | 340 | 165 | 278 | 356 | 241 | 170 | 142 | 111 | 269 | 95 | 312 | | 12 |
| TgCkBrAM17 | AM | Manacapuru | #7 | N. arch. | 289 | 242 | 203 | 156 | 340 | 165 | 278 | 356 | 243 | 170 | 142 | 111 | 269 | 91 | 312 | | 13 |
| TgCkBrAM18 | AM | Manacapuru | #7 | N. arch. | 289 | 242 | 203 | 156 | 340 | 165 | 278 | 356 | 243 | 170 | 142 | 111 | 269 | 91 | 312 | | 13 |
| TgCkBrAM19 | AM | Manacapuru | #7 | N. arch. | 289 | 242 | 203 | 156 | 340 | 165 | 278 | 356 | 237 | 170 | 142 | 111 | 271 | 95 | 312 | | 14 |
| TgCkBrAM20 | AM | Manacapuru | #7 | N. arch. | 289 | 242 | 203 | 156 | 340 | 165 | 278 | 356 | 235 | 170 | 142 | 111 | 271 | 95 | 312 | | 14 |
| TgCkBrAM21 | AM | Manacapuru | #7 | N. arch. | 289 | 242 | 203 | 156 | 340 | 165 | 278 | 356 | 237 | 170 | 142 | 111 | 271 | 95 | 312 | | 14 |
| TgCkBrAM22 | AM | Manacapuru | #7 | N. arch. | 289 | 242 | 203 | 156 | 340 | 165 | 278 | 356 | 239 | 170 | 142 | 111 | 269 | 95 | 312 | | 15 |
| TgCkBrRR6 | RR | Boa Vista | #7 | N. arch. | 289 | 242 | 203 | 156 | 340 | 165 | 278 | 356 | 239 | 170 | 142 | 111 | 269 | 95 | 312 | | 15 |
| TgCkBrAM23 | AM | Manacapuru | #7 | N. arch. | 289 | 242 | 203 | 156 | 340 | 165 | 278 | 356 | 235 | 170 | 142 | 111 | 269 | 95 | 312 | | 16 |
| TgCkBrAM24 | AM | Iranduba | #7 | N. arch. | 289 | 242 | 203 | 156 | 340 | 165 | 278 | 356 | 235 | 170 | 142 | 111 | 273 | 93 | 312 | | 17 |
| TgCkBrAM25 | AM | Iranduba | #7 | N. arch. | 289 | 242 | 203 | 156 | 340 | 165 | 278 | 356 | 235 | 170 | 142 | 111 | 273 | 93 | 312 | | 17 |
| TgCkBrAM26 | AM | Iranduba | #7 | N. arch. | 289 | 242 | 203 | 156 | 340 | 165 | 278 | 356 | 237 | 170 | 142 | 111 | 275 | 97 | 312 | | 18 |
| TgCkBrAM27 | AM | Iranduba | #7 | N. arch. | 289 | 242 | 203 | 156 | 340 | 165 | 278 | 356 | 231 | 170 | 142 | 111 | 269 | 95 | 312 | | 19 |
| TgCkBrRR1 | RR | Boa Vista | #7 | N. arch. | 289 | 242 | 203 | 156 | 340 | 165 | 278 | 356 | 231 | 170 | 142 | 111 | 261 | 95 | 312 | | 20 |
| TgCkBrRR2 | RR | Boa Vista | #7 | N. arch. | 289 | 242 | 203 | 156 | 340 | 165 | 278 | 356 | 231 | 170 | 142 | 111 | 261 | 95 | 312 | | 20 |
| TgCkBrRR3 | RR | Boa Vista | #7 | N. arch. | 289 | 242 | 203 | 156 | 340 | 165 | 278 | 356 | 231 | 170 | 142 | 111 | 261 | 95 | 312 | | 20 |
| TgCkBrRR4 | RR | Boa Vista | #7 | N. arch. | 289 | 242 | 203 | 156 | 340 | 165 | 278 | 356 | 231 | 170 | 142 | 111 | 261 | 95 | 312 | | 20 |
| TgCkBrRR5 | RR | Boa Vista | #7 | N. arch. | 289 | 242 | 203 | 156 | 340 | 165 | 278 | 356 | 237 | 170 | 142 | 111 | 269 | 95 | 312 | | 21 |
| TgCkBrAP2 | AP | Magazão | #7 | N. arch. | 289 | 242 | 203 | 156 | 340 | 165 | 278 | 356 | 241 | 170 | 142 | 111 | 273 | 95 | 312 | | 22 |
| TgCkBrAP3 | AP | Magazão | #7 | N. arch. | 289 | 242 | 203 | 156 | 340 | 165 | 278 | 356 | 241 | 170 | 142 | 111 | 273 | 95 | 312 | | 22 |

**Table 5.** (Continued)

| Isolate ID | State/Country[a] | Municipality | RFLP Genotype | MS Type[b] | Microsatellite markers[c,d] | | | | | | | | | | | | | | | N. of types | N. of subtypes |
|---|---|---|---|---|---|---|---|---|---|---|---|---|---|---|---|---|---|---|---|---|---|---|
| | | | | | TUB2 | W35 | TgM-A | B18 | B17 | M33 | IV.1 | XI.1 | M48 | M102 | N60 | N82 | AA | N61 | N83 | | |
| TgCkBrAP4 | AP | Mazagão | #7 | N. arch. | 289 | 242 | 203 | 156 | 340 | 165 | 278 | 356 | 239 | 170 | 142 | 111 | 271 | 95 | 312 | | 23 |
| TgCkBrAP6 | AP | Mazagão | #7 | N. arch. | 289 | 242 | 203 | 156 | 340 | 165 | 278 | 356 | 239 | 170 | 142 | 111 | 271 | 95 | 312 | | 23 |
| TgCkBrAP5 | AP | Mazagão | #7 | N. arch. | 289 | 242 | 203 | 156 | 340 | 165 | 278 | 356 | 233 | 170 | 142 | 111 | 269 | 95 | 312 | | 24 |
| TgCkBrAP21 | AP | Ferreira Gomes | #7 | N. arch. | 289 | 242 | 203 | 156 | 340 | 165 | 278 | 356 | 233 | 170 | 142 | 111 | 271 | 95 | 312 | | 25 |
| TgCkBrAP22 | AP | Ferreira Gomes | #7 | N. arch. | 289 | 242 | 203 | 156 | 340 | 165 | 278 | 356 | 227 | 170 | 142 | 111 | 271 | 91 | 312 | | 26 |
| TgCkBrAP19 | AP | Ferreira Gomes | #7 | N. arch. | 289 | 242 | 203 | 156 | 340 | 165 | 278 | 356 | 227 | 170 | 142 | 111 | 271 | 91 | 312 | | 26 |
| TgCkBrAP12 | AP | Tartarugalzinho | #71 | N. arch. | 289 | 242 | 203 | 156 | 340 | 165 | 278 | 356 | 211 | 170 | 140 | 111 | 263 | 91 | 312 | | 27 |
| TgCkBrAP17 | AP | Ferreira Gomes | #71 | N. arch. | 289 | 242 | 203 | 156 | 340 | 165 | 278 | 356 | 211 | 170 | 140 | 111 | 263 | 91 | 312 | | 27 |
| TgCkBrAP18 | AP | Ferreira Gomes | #71 | N. arch. | 289 | 242 | 203 | 156 | 340 | 165 | 278 | 356 | 211 | 170 | 140 | 111 | 263 | 89 | 312 | | 28 |
| TgCkBrPA2 | PA | Monte Alegre | #70 | N. arch. | 289 | 242 | 205 | 162 | 340 | 165 | 278 | 358 | 243 | 164 | 147 | 111 | 263 | 89 | 306 | 2 | 29 |
| TgCkBrPA5 | PA | Monte Alegre | #70 | N. arch. | 289 | 242 | 205 | 162 | 340 | 165 | 278 | 358 | 241 | 164 | 144 | 111 | 261 | 95 | 306 | | 30 |
| TgCkBrAM1 | AM | Parintins | #327[f] | N. arch. | 289 | 242 | 205 | 162 | 340 | 165 | 278 | 358 | 237 | 164 | 147 | 111 | 275 | 89 | 306 | | 31 |
| TgCkBrAM2 | AM | Parintins | #70 | N. arch. | 289 | 242 | 205 | 162 | 340 | 165 | 278 | 358 | 237 | 164 | 147 | 111 | 263 | 89 | 306 | | 32 |
| TgCkBrAM3 | AM | Parintins | #70 | N. arch. | 289 | 242 | 205 | 162 | 340 | 165 | 278 | 358 | 237 | 164 | 147 | 111 | 263 | 89 | 306 | | 32 |
| TgCkBrAM4 | AM | Parintins | #327[f] | N. arch. | 289 | 242 | 205 | 162 | 340 | 165 | 278 | 358 | 237 | 164 | 147 | 111 | 273 | 89 | 306 | | 33 |
| TgCkBrAM5 | AM | Parintins | #327[f] | N. arch. | 289 | 242 | 205 | 162 | 340 | 165 | 278 | 358 | 237 | 164 | 147 | 111 | 273 | 89 | 306 | | 33 |
| TgCkBrRR8 | RR | Bonfim | #277 | N. arch. | 289 | 242 | 205 | 162 | 340 | 165 | 278 | 358 | 213 | 164 | 142 | 111 | 265 | 91 | 306 | | 34 |
| TgCkBrAP9 | AP | Macapá | #70 | N. arch. | 289 | 242 | 205 | 162 | 340 | 165 | 278 | 358 | 237 | 164 | 149 | 111 | 267 | 89 | 306 | | 35 |
| TgCkBrAP15 | AP | Tartarugalzinho | #71 | N. arch. | 289 | 242 | 205 | 162 | 340 | 165 | 278 | 358 | 231 | 164 | 147 | 111 | 261 | 89 | 306 | | 36 |
| TgCkBrAP13 | AP | Tartarugalzinho | #28 | N. arch. | 291 | 248 | 209 | 160 | 344 | 165 | 278 | 358 | 209 | 190 | 138 | 125 | 265 | 85 | 304 | 3 | 37 |
| TgCkBrPA19 | PA | Soure | #28 | N. arch. | 291 | 248 | 209 | 160 | 344 | 165 | 278 | 358 | 209 | 190 | 138 | 129 | 265 | 83 | 304 | | 38 |
| TgCkBrAP7 | AP | Macapá | #28 | N. arch. | 291 | 248 | 209 | 160 | 344 | 165 | 278 | 358 | 209 | 190 | 138 | 133 | 269 | 83 | 304 | | 39 |
| TgCkBrAP8 | AP | Macapá | #28 | N. arch. | 291 | 248 | 209 | 160 | 344 | 165 | 278 | 358 | 209 | 190 | 138 | 133 | 269 | 83 | 304 | | 39 |
| TgCkBrAP25 | AP | Macapá | #28 | N. arch. | 291 | 248 | 209 | 160 | 344 | 165 | 278 | 358 | 209 | 190 | 138 | 135 | 265 | 83 | 304 | | 40 |
| TgCkBrPA20 | PA | Soure | #28 | N. arch. | 291 | 248 | 209 | 160 | 344 | 165 | 278 | 358 | 209 | 190 | 138 | 137 | 265 | 83 | 304 | | 41 |
| TgCkBrAP11 | AP | Itaubal | #28 | N. arch. | 291 | 248 | 209 | 160 | 344 | 165 | 278 | 358 | 209 | 190 | 138 | 141 | 265 | 83 | 304 | | 42 |
| TgCkBrPA21 | PA | Soure | #326[f] | N. arch. | 291 | 248 | 209 | 160 | 344 | 165 | 278 | 358 | 209 | 166 | 140 | 113 | 279 | 87 | 306 | | 43 |
| TgCkBrPA22 | PA | Soure | #326[f] | N. arch. | 291 | 248 | 209 | 160 | 344 | 165 | 278 | 358 | 209 | 166 | 140 | 113 | 279 | 87 | 306 | | 43 |
| TgCkBrPA18 | PA | Soure | #325[f] | N. arch. | 291 | 248 | 209 | 160 | 344 | 165 | 278 | 358 | 227 | 190 | 138 | 111 | 265 | 87 | 306 | | 44 |
| TgCkBrAM9 | AM | Tabatinga | #14 | N. arch. | 289 | 242 | 205 | 162 | 340 | 165 | 278 | 356 | 213 | 164 | 147 | 111 | 265 | 95 | 316 | 4 | 45 |
| TgCkBrAM13 | AM | Tabatinga | #14 | N. arch. | 289 | 242 | 205 | 162 | 340 | 165 | 278 | 356 | 213 | 164 | 147 | 111 | 265 | 97 | 316 | | 46 |
| TgCkBrAM12 | AM | Tabatinga | #14 | N. arch. | 289 | 242 | 205 | 162 | 340 | 165 | 278 | 356 | 213 | 164 | 147 | 111 | 265 | 97 | 316 | | 46 |
| TgCkBrAM11 | AM | Tabatinga | #14 | N. arch. | 289 | 242 | 205 | 162 | 340 | 165 | 278 | 356 | 213 | 164 | 147 | 111 | 263 | 87 | 316 | | 47 |
| TgCkBrAM14 | AM | Tabatinga | #14 | N. arch. | 289 | 242 | 205 | 162 | 340 | 165 | 278 | 356 | 213 | 164 | 147 | 111 | 263 | 87 | 316 | | 47 |
| TgCkBrAM15 | AM | Tabatinga | #14 | N. arch. | 289 | 242 | 205 | 162 | 340 | 165 | 278 | 356 | 213 | 164 | 147 | 111 | 263 | 87 | 316 | | 47 |
| TgCkBrAP20 | AP | Ferreira Gomes | #334[f] | N. arch. | 289 | 242 | 205 | 162 | 340 | 165 | 278 | 356 | 233 | 164 | 145 | 111 | 271 | 95 | 312 | | 48 |
| TgCkBrAC14 | AC | Epitaciolândia | #194 | N. arch. | 291 | 246 | 207 | 156 | 344 | 169 | 278 | 356 | 223 | 168 | 140 | 111 | 277 | 83 | 308 | 5 | 49 |
| TgCkBrAC15 | AC | Epitaciolândia | #194 | N. arch. | 291 | 246 | 207 | 156 | 344 | 169 | 278 | 356 | 223 | 168 | 140 | 111 | 277 | 83 | 308 | | 49 |

*(Continued)*

**Table 5.** (Continued)

| Isolate ID | State/Country[a] | Municipality | RFLP Genotype | MS Type[b] | Microsatellite markers[c,d] | | | | | | | | | | | | | | | N. of types | N. of subtypes |
|---|---|---|---|---|---|---|---|---|---|---|---|---|---|---|---|---|---|---|---|---|---|---|
| | | | | | TUB2 | W35 | TgM-A | B18 | B17 | M33 | IV.1 | XI.1 | M48 | M102 | N60 | N82 | AA | N61 | N83 | | |
| TgCkBrAC16 | AC | Epitaciolândia | #194 | N. arch. | 291 | 246 | 207 | 156 | 344 | 169 | 278 | 356 | 223 | 168 | 140 | 111 | 277 | 83 | 308 | | 49 |
| TgCkBrAM28 | AM | Iranduba | #8 (BrIII) | N. arch. | 289 | 242 | 205 | 160 | 348 | 165 | 278 | 356 | 213 | 190 | 142 | 111 | 263 | 107 | 312 | 6 | 50 |
| TgCkBrAC18 | AC | Assis Brasil | #8 (BrIII) | N. arch. | 289 | 242 | 205 | 160 | 348 | 165 | 278 | 356 | 213 | 190 | 142 | 113 | 261 | 101 | 312 | | 51 |
| TgCkBrAC12 | AC | Sena Madureira | #97 | N. arch. | 291 | 246 | 209 | 158 | 344 | 173 | 274 | 356 | 217 | 180 | 142 | 109 | 263 | 85 | 315 | 7 | 52 |
| TgCkBrAC13 | AC | Sena Madureira | #97 | N. arch. | 291 | 246 | 209 | 158 | 344 | 173 | 274 | 356 | 217 | 180 | 142 | 109 | 263 | 85 | 315 | | 52 |
| TgCkBrAP23 | AP | Porto Grande | #109 | N. arch. | 291 | 242 | 207 | 164 | 334 | 165 | 278 | 356 | 227 | 168 | 147 | 105 | 277 | 97 | 310 | 8 | 53 |
| TgCkBrAP24 | AP | Porto Grande | #109 | N. arch. | 291 | 242 | 207 | 164 | 334 | 165 | 278 | 356 | 227 | 168 | 147 | 105 | 277 | 97 | 310 | | 53 |
| TgCkBrAC8 | AC | Acrelândia | #219 | N. arch. | 289 | 242 | 205 | 162 | 344 | 165 | 278 | 356 | 235 | 164 | 147 | 111 | 271 | 89 | 316 | 9 | 54 |
| TgCkBrAC9 | AC | Acrelândia | #219 | N. arch. | 289 | 242 | 205 | 162 | 344 | 165 | 278 | 356 | 235 | 164 | 147 | 111 | 271 | 89 | 316 | | 54 |
| TgCkBrAP10 | AP | Itaubal | #332[f] | N. arch. | 291 | 248 | 209 | 160 | 344 | 169 | 278 | 356 | 209 | 166 | 142 | 121 | 293 | 93 | 308 | 10 | 55 |
| TgCkBrAP14 | AP | Tartarugalzinho | #333[f] | N. arch. | 291 | 248 | 209 | 160 | 344 | 169 | 278 | 356 | 209 | 166 | 142 | 121 | 293 | 93 | 308 | | 55 |
| TgCkBrAC2 | AC | Rio Branco | #336[f] | N. arch. | 291 | 242 | 205 | 160 | 348 | 169 | 278 | 358 | 239 | 192 | 136 | 111 | 263 | 113 | 306 | 11 | 56 |
| TgCkBrAC3 | AC | Rio Branco | #336[f] | N. arch. | 291 | 242 | 205 | 160 | 348 | 169 | 278 | 358 | 239 | 192 | 136 | 111 | 263 | 113 | 306 | | 56 |
| TgCkBrAC10 | AC | Sena Madureira | #11 (BrII) | S.A.1 | 289 | 242 | 205 | 160 | 342 | 165 | 278 | 358 | 237 | 164 | 145 | 111 | 314 | 89 | 308 | 12 | 57 |
| TgCkBrAC7 | AC | Acrelândia | #19 | S.A.4 | 291 | 242 | 205 | 160 | 362 | 165 | 278 | 356 | 237 | 174 | 140 | 111 | 265 | 91 | 314 | 13 | 58 |
| TgCkBrAC4 | AC | Plácido de Castro | #19 | N. arch. | 291 | 242 | 205 | 160 | 364 | 165 | 278 | 356 | 235 | 174 | 140 | 111 | 265 | 97 | 314 | 14 | 59 |
| TgCkBrAC6 | AC | Acrelândia | #41 | N. arch. | 291 | 242 | 205 | 160 | 342 | 165 | 278 | 354 | 225 | 166 | 140 | 111 | 271 | 91 | 308 | 15 | 60 |
| TgCkBrAC5 | AC | Plácido de Castro | #57 | N. arch. | 289 | 242 | 207 | 160 | 348 | 169 | 278 | 356 | 213 | 196 | 147 | 111 | 269 | 87 | 308 | 16 | 61 |
| TgCkBrPA23 | PA | Soure | #95 | N. arch. | 289 | 248 | 205 | 164 | 336 | 165 | 274 | 356 | 215 | 166 | 149 | 113 | 265 | 95 | 325 | 17 | 62 |
| TgCkBrAP16 | AP | Tartarugalzinho | #97 | N. arch. | 291 | 242 | 207 | 164 | 340 | 167 | 272 | 356 | 213 | 174 | 164 | 113 | 213 | 87 | 312 | 18 | 63 |
| TgCkBrAP1 | AP | Magazão | #98 | N. arch. | 289 | 242 | 207 | 162 | 336 | 171 | 272 | 354 | 223 | 172 | 140 | 127 | 261 | 85 | 330 | 19 | 64 |
| TgCkBrPA17 | PA | Soure | #258 | N. arch. | 291 | 242 | 207 | 160 | 340 | 167 | 274 | 356 | 231 | 176 | 162 | 125 | 265 | 99 | 324 | 20 | 65 |
| TgCkBrPA9 | PA | Monte Alegre | #324[f] | N. arch. | 293 | 242 | 207 | 160 | 344 | 167 | 278 | 356 | 243 | 174 | 134 | 109 | 265 | 97 | 331 | 21 | 66 |
| TgCkBrAM6 | AM | Parintins | #328[f] | N. arch. | 289 | 242 | 203 | 160 | 340 | 165 | 278 | 356 | 213 | 174 | 140 | 111 | 275 | 93 | 316 | 22 | 67 |
| TgCkBrAM10 | AM | Tabatinga | #329[f] | N. arch. | 291 | 244 | 203 | 156 | 344 | 165 | 278 | 356 | 235 | 186 | 138 | 115 | 265 | 87 | 316 | 23 | 68 |
| TgCkBrAM16 | AM | Tabatinga | #330[f] | N. arch. | 289 | 248 | 205 | 160 | 342 | 165 | 278 | 358 | 211 | 166 | 142 | 121 | 281 | 89 | 306 | 24 | 69 |
| TgCkBrAM29 | AM | Rio Preto da Eva | #331[f] | N. arch. | 291 | 244 | 203 | 158 | 344 | 167 | 274 | 356 | 213 | 186 | 140 | 109 | 259 | 117 | 317 | 25 | 70 |
| TgCkBrAC1 | AC | Rio Branco | #335[f] | N. arch. | 289 | 246 | 203 | 158 | 350 | 165 | 274 | 356 | 217 | 168 | 142 | 107 | 307 | 83 | 318 | 26 | 71 |
| TgCkBrAC11 | AC | Sena Madureira | #337[f] | N. arch. | 289 | 240 | 203 | 158 | 342 | 165 | 276 | 356 | 207 | 174 | 136 | 109 | 261 | 85 | 310 | 27 | 72 |
| TgCkBrAC17 | AC | Epitaciolândia | #338[f] | N. arch. | 289 | 246 | 203 | 164 | 342 | 169 | 274 | 356 | 225 | 170 | 140 | 129 | 285 | 95 | 312 | 28 | 73 |
| TgCkBrRR7 | RR | Boa Vista | #339[f] | N. arch. | 293 | 242 | 203 | 162 | 350 | 169 | 274 | 356 | 209 | 176 | 136 | 113 | 283 | 95 | 306 | 29 | 74 |
| TgCkGy37 | Gy | Lethem | #12 | N. arch. | 291 | 242 | 205 | 160 | 336 | 165 | 278 | 356 | 211 | 190 | 142 | 109 | 267 | 89 | 312 | 30 | 75 |
| TgCkGy39 | Gy | Lethem | #12 | N. arch. | 291 | 242 | 205 | 160 | 336 | 165 | 278 | 356 | 211 | 190 | 142 | 109 | 267 | 89 | 312 | | 75 |
| TgCkGy40 | Gy | Lethem | #12 | N. arch. | 291 | 242 | 205 | 160 | 336 | 165 | 278 | 356 | 211 | 190 | 142 | 109 | 267 | 89 | 312 | | 75 |
| TgCkGy38 | Gy | Lethem | #343[f] | N. arch. | 289 | 242 | 203 | 162 | 342 | 167 | 272 | 356 | 219 | 168 | 142 | 117 | 195 | 99 | 314 | 31 | 76 |
| TgCkVe14 | Ve | Sta Elena de Uairén | #48 | N. arch. | 289 | 242 | 205 | 164 | 344 | 165 | 278 | 358 | 221 | 172 | 140 | 111 | 265 | 87 | 312 | 32 | 77 |
| TgCkVe15 | Ve | Sta Elena de Uairén | #48 | N. arch. | 289 | 242 | 205 | 164 | 344 | 165 | 278 | 358 | 221 | 172 | 140 | 111 | 265 | 87 | 312 | | 77 |

**Table 5.** (Continued)

| Isolate ID | State/ Country[a] | Municipality | RFLP Genotype | MS Type[b] | Microsatellite markers[c,d] | | | | | | | | | | | | | | | N. of types | N. of subtypes |
| --- | --- | --- | --- | --- | --- | --- | --- | --- | --- | --- | --- | --- | --- | --- | --- | --- | --- | --- | --- | --- | --- |
| | | | | | TUB2 | W35 | TgM-A | B18 | B17 | M33 | IV.1 | XI.1 | M48 | M102 | N60 | N82 | AA | N61 | N83 | | |
| TgCkVe16 | Ve | Sta Elena de Uairén | #123 | Type III | 289 | 242 | 205 | 160 | 336 | 165 | 278 | 356 | 213 | 174 | 140 | 111 | 267 | 87 | 316 | 33 | 78 |
| TgCkBo1 | Bo | Villa Busch | #95 | N. arch. | 291 | 242 | **203** | **164** | **344** | **167** | **276** | 356 | 209 | 178 | 138 | 111 | 261 | 103 | 306 | 34 | 79 |
| TgCkBo2 | Bo | Villa Busch | #340[f] | N. arch. | 291 | 242 | 205 | 160 | **340** | 165 | 278 | 356 | 213 | 164 | 140 | 111 | 279 | 89 | 308 | 35 | 80 |
| TgCkPe11 | Pe | Inãpari | #341[f] | N. arch. | 291 | **246** | 207 | 160 | **344** | 169 | 278 | 356 | 217 | 170 | 140 | 113 | 291 | 83 | 308 | 36 | 81 |
| TgCkPe12 | Pe | Inãpari | #341[f] | N. arch. | 291 | **246** | 207 | 160 | **344** | 169 | 278 | 356 | 217 | 170 | 140 | 113 | 291 | 83 | 308 | | 81 |
| TgCkPe13 | Pe | Inãpari | #341[f] | N. arch. | 291 | **246** | 207 | 160 | **344** | 169 | 278 | 356 | 219 | 170 | 140 | 113 | 291 | 83 | 308 | 37 | 82 |
| TgCkPe14 | Pe | Noaya | #342[f] | N. arch. | 291 | 244 | 207 | **168** | **340** | **171** | 274 | 356 | 211 | 174 | 138 | 115 | 263 | 87 | 312 | 38 | 83 |

[a] AC = Acre state; AM = Amazonas state; AP = Amapá state; PA = Pará state; RR = Roraima state; Gy = Guyana; Ve = Venezuela; Bo = Bolivia; Pe = Peru.

[b] N. arch. = Non-archetypal MS type; S.A.1 = South American 1; S.A.4 = South American 4.

[c] Microsatellite markers in bold italics are typing markers and the others are fingerprinting markers.

[d] Alleles in bold are unique/non-archetypal ones.

[e] ENT, ME49 and NED are archetypal reference strains from goat (USA), sheep (USA) and human (France), respectively.

[f] new RFLP genotypes.

A phylogeographic study of *T. gondii* suggested the South American origin of the parasite at 1.5 million years ago following the arrival of felids in this area of the world [72]. Therefore, this ancient history in an environment with favorable climatic conditions and the presence of many species of wild felids and prey circulating in a vast territory may have contributed to the high genetic diversity of *T. gondii* in South America involving the spread and survival of oocysts and sexual recombination through carnivorism [4,72].

Brazil, Colombia, and Argentina are hotspots for *T. gondii* genetic diversity in South America. In Brazil, this knowledge comes particularly from isolates obtained from the Southeastern region; however, when describing the circulation of *T. gondii* genotypes in the country, it is important to note that Brazil is the fifth largest country in the world and the largest country in South America and Latin America occupying almost 50% of South American territory; São Paulo city, in the Southeastern region is separated from Manaus city, in the Northern region, by almost 4,000 km. Additionally, the human population is not evenly distributed across the country, with high concentrations in the Southeast and along the coast. Therefore, most of the information regarding *T. gondii* diversity in Brazil has been focused on the most populated and urbanized areas. To fill this gap, the largest collection of *T. gondii* isolates (116 isolates from chickens) was obtained from the Amazon region that is the most important area of the remaining forest in South America. Additionally, this is the largest collection of new *T. gondii* PCR-RFLP genotypes recently described in a single study (20 new genotypes), thus contributing to the knowledge on the genetic diversity of *T. gondii* worldwide. On the other hand, in the present study, the wild animal population could not be assessed because of the limitations of resources, legal permissions, logistic in extremely remote areas, and, particularly, safety, so it is almost certain that the present study could not still picture the complete diversity of alleles circulating in the Amazon region, because it is known that intermediate and definitive host diversity in an environment can contribute significantly to the diversification of the parasite through the establishing of a wild cycle [4], as demonstrated in French Guiana [8].

Bioassays of free-range chicken tissues (brain and heart) in mice were highly successful in obtaining *T. gondii* isolates for biological and molecular studies, showing the advantage of using seropositive chickens with MAT titers ≥ 20 [70]. The frequency of anti-*T. gondii* antibodies in the examined free-range chickens was high (68.1%), and this was in agreement with data reported in recent studies in Brazil where levels were 71.3% (77/108) in Minas Gerais state [52], 49.2% (294/597) in Rio Grande do Sul state [73], 38.8% (198/510) in Espírito Santo state [37], and 36.0% (72/200) in Alagoas state [74], thus corroborating the high circulation of the parasite and the importance of epidemiologic studies to establish preventive measures and sanitary education.

Using Mn-nPCR-RFLP, high diversity was observed among the isolates, with 42 non-archetypal genotypes being reported. Previously, there was no information about the *T. gondii* genotypes in the Brazilian states of Amapá, Roraima, and Acre or in Bolivia. In the Northern Brazilian states of Rondônia and Pará, 25 genotypes corresponding to 47 strains have been previously reported [6,61] with only six genotypes in common with the present study (#7, Type BrIII (#8), #28, #41, #70, and #258). The Brazilian clonal Type BrI (#6) and BrIII (#8) lineages had already been reported in the Northern region of Brazil, and the present study reported Type BrII, thus confirming the high circulation of these most prevalent genotypes in the country. Genotype #7 was the most frequent in the present study with a large distribution in four of the Brazilian states searched, and it could represent a regional clonal lineage that is closely related to the archetypal Type III (#2) lineage and has also been reported as a dominant lineage in Central America and other South American countries, including Argentina and Guyana [10,30].

Archetypal clonal Types I (#10), II (#1 and #3), and III (#2) were not observed. These classical types exhibit low frequencies in the Brazilian territory, with a small number of Type I and Type II strains circulating in the Southern region of Brazil [75–78], Type II variant (#3) particularly on Fernando de Noronha Island [6,79], and some Type III strains were isolated in all regions of Brazil except for the North [6,30]. In South America, Type III has already been reported in Guyana, Peru, Argentina, and Chile, but it appears to be more frequent in Central America. Type II has been reported in Argentina and particularly in Chile, where its geographic isolation due to the Andes Mountains could play a role in this predominance [6,10,30]. Studies on global diversity of *T. gondii* has showed that the world trade, human migration and slave trade through maritime route had a remarkable role in the spread of the archetypal genotypes from Europe to other continents during colonial time through transportation of invasive mice, rats and cats [4,71,80]. So, in Brazil and other countries of South America this event has probably occurred between 16th and 19th centuries with the territory colonization by Portugal and Spain, respectively. However, in an environment already populated with hundreds of strains, there was no space for expansion, probably explaining why archetypal genotypes are so limited in those territories. In Brazil, the colonization started in the coast, particularly in the South and Southeast, so this would have preserved the Northern region from contact with these new strains because of the geographical distance together with undeveloped means of terrestrial transportation.

Samples from Guyana, Venezuela, Bolivia, and Peru were collected from municipalities bordering Brazil; however, the eight genotypes reported here (including four new genotypes) were different from those from the five Brazilian states searched (Fig 3). Three of them (#12, #95, and #123) have never been reported in Brazil. Genotype #12 has been previously reported as the most prevalent genotype in Guyana [6,81], and it was also observed in the present study, likely indicating its high circulation in this country. These differences may just reflect both the high diversity in the vast biome studied, where borders are merely administrative separation, and the limited number of samples collected. Although few isolates were analyzed in the present study, the results contribute to our knowledge of the distribution and diversity of *T. gondii* genotypes in other South American countries.

Phenotypically, *T. gondii* isolates in mice are usually classified as virulent, non-virulent, or of intermediate virulence. This classification can be defined in a second round of mouse (normally Swiss or Balb-c lineages) inoculations (normally intraperitoneal), after isolation, based on the percentage of mortality of mice infected with different parasite (normally tachyzoites) doses (usually $10^0$, $10^1$, $10^2$, $10^3$) [82]. This system can be very expensive, time and space consuming when applied to a high number of samples as in the present study. Alternatively, although less reliable, virulence could be defined during the isolation process, also based on mouse mortality, when unknow doses of parasites (normally bradyzoites from digested tissues from chronic host infections) are inoculated (generally subcutaneously) in mice, as in most of the isolation studies in Brazil. The cumulative mortality parameter to define virulence in mice can be considered ethically questionable nowadays. In the present report, virulence definition was based on other criteria, using an endpoint for euthanasia [27]; other studies in Brazil also adopted endpoints to mouse euthanasia [83,84]. Most isolates exhibited a virulent phenotype in mice, as expected for isolates from South America [85]. It is important to remark that many factors can interfere with mouse virulence including the stage of the parasite inoculated, route, dose, lineages of mice used in the bioassays and the strain of the parasite [1]. The number of mice inoculated in each bioassay group could also interfere with virulence interpretation, particularly if only one mouse gets infected as occurred in some groups in the present study. In summary, there is no standardization on mice virulence assays in the current literature, making result comparisons difficult.

Molecular markers could be an alternative to predict *T. gondii* virulence in mice, minimizing the use of this model. In this sense, the CS3 marker was used in the present study to assess virulence in a mouse model, as it was previously reported as strongly linked to acute virulence in this model [86]. It was observed that CS3 allelles I and II had a strong association with virulent and intermediate virulent isolates and CS3 allele III with avirulent isolates. It is necessary to be cautious when comparing these results with others because, as exposed above, there is a lack of standardization in the isolation methods related to the route of inoculation (subcutaneous or intraperitoneal), mouse lineage (Balb-c or Swiss), dose (protocols with known doses and unknow doses), stage of the parasite inoculated (bradyzoites or tachyzoites) and criteria for virulence, all of which are factors that could interfere with the results.

Results of *T. gondii* mouse virulence based in the CS3 I, II and III alleles in the present study are comparable to those from Brazilian isolates (a total of 194 isolates from diverse hosts analyzed) reported in other studies [9,38,50,55,61,62,77,87–90]. For other studies from Brazil (a total of 179 isolates from diverse hosts analyzed), there is a major concordance for CS3 alleles I and II as virulent or intermediate but CS3 allele III showed avirulent, virulent or intermediate virulences [40,55,83,84,91,92]. Also, two studies from Spain showed that three clonal Type III isolates (CS3 = III) were mouse virulent [93,94]. It has been suggested that factors such as different proteins (e.g., pseudokinases, kinases, polymorphic dense granule protein) secreted into the host cell during invasion would have a role in the virulence of isolates with CS3 allele III [40]. It is noteworthy that nine clonal Type II (CS3 = II) sheep isolates from Spain were reported as avirulent for mice [94], corroborating previous studies in Brazil [40,60,78,87], where this archetypal genotype has a very low circulation, which suggests a different modulation of virulence for this genotype. So, overall, the present study, and most of the previous studies in Brazil suggests that, when examining a set of samples, isolates with CS3 alleles I and II have great probability to be virulent or intermediately virulent in mice, but in Europe, where archetypal type II predominates, it seems this conclusion does not apply. It is important to point out that no association of CS3 allele type with clinical manifestations related to human congenital toxoplasmosis was observed [40].

More recently, it has been demonstrated that ROP18 and ROP5 gene alleles could predict *T. gondii* virulence in mice [95], besides it was observed that strains with the ROP18 type I allele are associated with severe ocular inflammation in ocular toxoplasmosis in Colombia [96], but more studies are necessary to corroborate this kind of association. The definitive role of molecular markers in determining the exact virulence in mice or in clinical human manifestations of toxoplasmosis has yet to be determined.

Diversity analysis revealed a high *D* value (0.872) for the 116 isolates studied here, but the Southeastern region of Brazil exhibited a much higher value (0.961) that was likely reflective of the impact of the predominance of the #7 genotype (40 isolates) in the Amazon region. When strains from other studies in Northern Brazil were included in the analysis (SI 2), the *D* value increased to 0.908.

The MS genotyping of *T. gondii* isolates is based on two discriminatory levels that include typing and fingerprinting levels [31]. At the typing level, 37 non-archetypal and one archetypal Type III were observed in the present report, and this level is also important for searching for non-archetypal/unique alleles. The fingerprinting level aids epidemiologic studies in cases of toxoplasmosis outbreaks and laboratory contamination, as it is highly discriminatory. In the present study, these 38 types were differentiated into 83 genotypes, thus emphasizing the high diversity in the studied region.

Many of the genotypes reported here have been previously associated with cases of human toxoplasmosis in Brazil including TypeBrIII (#8), TypeBrII (#11), #14, #41, and #71 [41,49,83]. These results likely reflect the circulation of the strains in the region where the

cases occurred. For example, Types BrIII (#8) and BrII (#11) are among the most prevalent genotypes circulating in Brazil. Carneiro et al. [41] observed that 62.5% (15/24) of congenital toxoplasmosis cases were caused by strains with genotypes that had already been identified in animal hosts.

Non-archetypal strains with many unique MS polymorphisms in many typing markers (three–six) have been associated with severe disseminated human toxoplasmosis in immunocompetent patients in French Guiana and Suriname [15,31,97], and this is likely related to a wild cycle. Allele 203 at the TgM-A marker appears to be only present in South America [16], and it was one of the non-archetypal alleles in four toxoplasmosis cases, where three of which combined with the W35-246 allele. Interestingly, in the case of rare pulmonary toxoplasmosis in an immunocompetent patient from São Paulo state in Southeastern Brazil, the isolate involved also had a combination of alleles 203 and 246 at markers TgM-A and W35, respectively [19,27]. In the present report, the TgM-A 203 allele was present in nine MS types, and W35 246 allele in six MS types. Both were present in two isolates. When examining a limited number of isolates (222 isolates) from different regions from Brazil that were previously genotyped by MS (S2 Table), it was observed that these alleles are very rare in the population. Most isolates from the present report possessed 3–5 unique MS alleles, whereas most isolates from the Northeastern and Southeastern region possessed 0-2 unique MS alleles.

The mechanisms associated with the outcome of human disease are complex. These studies were recently revised [4,98] and involve human immune status, individual genetic background and adaptation, exposure rate, and parasite genetics where type I or non-archetypal alleles are more highly associated with severe toxoplasmosis. It is important to note that it is unknown which *T. gondii* genotypes are present in the infected asymptomatic population, and this population represents most cases.

The phylogenetic network constructed revealed a population structured into five groups (Fig 5), and although samples from the Amazon region and from the Southeastern region can be observed in all groups, there is a clear divergence between the two major groups. Most of the genotypes from the Amazon region are clustered in Group 1, and most of genotypes from the Southeastern Brazil are clustered in Group 5 together with Type BrI (#6) and Type BrII (#11) Brazilian clonal lineages. In addition, the populations are different between Acre and the other three Brazilian states in Amazon region (Amapá, Amazonas, Pará) (Fig 3). These are indication of geographical separation of *T. gondii* population in the Amazon region versus the Southeastern region of Brazil, as well as spatial diversity within the Amazon region. These results corroborate previous results suggesting the presence of an Amazonic phylogenetic branch separated from the archetypal lineages [61]. The most prevalent genotype observed in the present study, genotype #7, clustered with the Type III (#2) and Type BrIII (#8) lineages, thus corroborating a previous study [10] that had already reported this proximity when examining strains from South America and Central America. The structurally diverging strains from the Amazon region and the Southeastern Brazil are likely related to the historic evolution of the parasite in South America, geographic distance, different colonization processes, and migratory waves.

From the perspective of One Health, the importance of these findings for the population living in the region has been observed in studies with human populations in urban and rural areas [21,99] and in some studies also describing outbreaks of clinical toxoplasmosis [100,101]. Also in the Amazon region, in French Guiana and Suriname, description of very severe clinical toxoplasmosis, even in immunocompetent patients, were described, and called "Amazonian Toxoplasmosis" [12,17]. However, there are still few studies when observing the vast area that makes up the Amazon territory and the great diversity of cultures and wildlife that live in this region, which makes it difficult to implement control measures, whether

animal or environmental, but point to the need to implement strategic measures with the human populations.

The present study reports the large distribution of *T. gondii* in the Amazon region, contributes to the knowledge of the diversity of the parasite worldwide and reveals a major divergence between strains from the Amazon region, that is less urbanized and populated, and strains from Southeast Brazil which is the most populated and urbanized Brazilian region.

## Supporting information

**S1 Table. Data on chicken samples from Amazon region.**
(XLSX)

**S2 Table. Microsatellite genotyping of previously studied *Toxoplasma gondii* strains from Brazil.**
(XLSX)

**S3 Table. PCR-RFLP genotypes used to construct a NeighborNet phylogenetic network of *Toxoplasma gondii*.**
(XLSX)

## Acknowledgements

The MAT antigen was kindly provided by Dr. J. P. Dubey of the Laboratory of Animal Parasitic Diseases, United States Department of Agriculture (USDA), Beltsville, Maryland, USA. We thank Renato Caravieri for his technical assistance. We are grateful to Daniel Ajzenberg, Helder Batista, Daniel Herreira, Fábio Albuquerque, and Thiago Moreira for their assistance during chicken sampling.

## Author contributions

**Conceptualization:** Solange M. Gennari.

**Data curation:** Solange M. Gennari, Hilda F. J. Pena, Herbert S. Soares.

**Formal analysis:** Hilda F. J. Pena, Ricardo A. Dias.

**Funding acquisition:** Solange M. Gennari, Hilda F. J. Pena.

**Investigation:** Hilda F. J. Pena, Herbert S. Soares, Antonio H. H. Minervino, Francisco F. V. de Assis, Bruna F. Alves, Solange Oliveira, Juliana Aizawa.

**Methodology:** Solange M. Gennari, Hilda F. J. Pena.

**Project administration:** Solange M. Gennari, Hilda F. J. Pena.

**Resources:** Hilda F. J. Pena, Antonio H. H. Minervino, Francisco F. V. de Assis, Ricardo A. Dias.

**Software:** Hilda F. J. Pena, Ricardo A. Dias, Chunlei Su.

**Supervision:** Solange M. Gennari, Hilda F. J. Pena.

**Validation:** Solange M. Gennari, Hilda F. J. Pena.

**Visualization:** Solange M. Gennari.

**Writing – original draft:** Solange M. Gennari.

**Writing – review & editing:** Solange M. Gennari, Hilda F. J. Pena, Bruna F. Alves, Chunlei Su.

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
