## [Decision Letter · Decision Letter 0]

3 Sep 2024

Dear Dr. Pena,

Thank you very much for submitting your manuscript "**A population study of *Toxoplasma gondii* in the Amazon region expands current knowledge of the genetic diversity in South America**" for consideration at PLOS Neglected Tropical Diseases. As with all papers reviewed by the journal, your manuscript was reviewed by members of the editorial board and by several independent reviewers. In light of the reviews (below this email), we would like to invite the resubmission of a significantly-revised version that takes into account the reviewers' comments. 

We cannot make any decision about publication until we have seen the revised manuscript and your response to the reviewers' comments. Your revised manuscript is also likely to be sent to reviewers for further evaluation.

Sincerely,

Hamed Kalani

Academic Editor

Hira Nakhasi

Section Editor

Reviewer #1: 

- Were possible cross reactions taken into account when performing serological assays? Please expand on this. 

- Can authors account for the differences between the number of seropositive chickens and the final number of animals bioassayed in mice? As per the methodology description, tissues from all positive chicken were bioassayed in mice, however the number of bioassays is lower than the seropositive animals. This should be explained in the methodology or results section. 

- L505: Recent studies have shown that the CS3 marker is actually not a good predictor of Toxoplasma virulence in mice (e.g. doi: 10.1186/s13567-021-00953-7). This should be acknowledged and addressed in the discussion.

Reviewer #2: 

There a few points relating to the murine infections that were not clear:

1) Since the starting dose of T. gondii is unknown is it possible that the difference in virulence is not due to the isolate but due to initial dose? If this is the case could the authors comment on this possibility in the discussion.

2) In the Materials and Methods it states:

The inoculated mice were considered “infected” with the parasite when tachyzoites or tissue cysts were present in the lungs and brain, respectively. (Lines 159-161). I just wanted to confirm that this is what “infected” means in Table 3 e.g. for TgCkBrAC3 where it says Sick/Infected 0/3 all three mice had tachyzoites/tissue cysts but did not display sickness behaviour? If this isn’t what “infected” means here could the authors please clarify.

3) Potentially this is me not understanding the table correctly but for some strains there is only 1 mouse sick out of 1 mouse infected, whereas others have 3. If only 1 mouse is infected is it really possible to identify intermediate virulence? Again, I think it could be commented upon in the discussion that this is based on very small numbers and unequal group sizes so predictions of virulence should be made with caution.

Reviewer #3: 

1.1-Line 348 “:.... genotypes in Brazil were identified for the first time in the Brazilian Amazonr”. Review Amazon r?

Lines 352- 355: “These genotypes included 81genotypes (from 218 isolates) previously reported in the Southeastern region of Brazil, 52 genotypes (116 isolates from this study and 19 from previous studies) in the Northern region of Brazil”. Wich previous studies are you referring? Please cite the reference(s).

1.2-Lines 361- 365: “Most of the genotypes observed 362 in the Amazon region in this study and from other studies in the Northern region of Brazil were clustered in Group 1 (29 genotypes, 44 isolates), and most of the genotypes described in the Southeastern region of Brazil were clustered in Group 5 (51 genotypes, 230 isolates)”. Also, include the referrence (s) related to “from other studies”.

1.3-Lines 418-419:“.... (103 isolates from 419 the present study and 47 isolates from previous studies.....” Change to ....103 isolates from 419 from the present study and 47 isolates from previous studies...”. 

2.1-Lines 438-446 Brazil, Colombia, and Argentina are hotspots for T. gondii genetic diversity 437 in South America. In Brazil, this knowledge comes particularly from isolates obtained from the Southeastern region; however, when describing the circulation of T. gondii genotypes in the country, it is important to note that Brazil is the fifth largest country in the world and the largest country in South America and Latin America occupying almost 50% of South American territory; São Paulo city, in the Southeastern region is separated from Manaus city, in the Northern region, by almost 4,000 km. Additionally, the human population is not evenly distributedacross the country, with high concentrations in the Southeast and along the coast. Therefore, most of the information regarding T. gondii diversity in Brazil has been focused on the most populated and urbanized areas”. 

Discussion suggested: From the perspective of One Health, what would be the impact of these findings for the environment and the population living in the studied áreas?

2.2-Lines 479-484 –“ Archetypal clonal Types I (#10), II (#1 and #3), and III (#2) were not observed. These classical types exhibit low frequencies in the Brazilian territory, with a small number of Type I and Type II strains circulating in the Southern region of Brazil [42–45], Type II variant (#3) particularly on Fernando de Noronha Island [3,46], and some Type III strains were isolated in all regions of Brazil except for the North [3,29]”. 

Discussion: How do the authors explain the differences in prevalence and distribution of these strains found in the North versus Southern region of Brazil?

2.3- Lines 489 – 498: “Samples from Guyana, Venezuela, Bolivia, and Peru were collected from municipalities bordering Brazil; however, the eight genotypes reported here (including four new genotypes) were different from those from the five Brazilian states searched (Fig 3). Three of them (#12, #95, and #123) have never been reported in Brazil. Genotype #12 has been previously reported as the most prevalent genotype in Guyana [3,37], and it was also observed in the present study, likely indicating its high circulation in this country. Although few isolates were analyzed in the present study, the results contribute to our knowledge of the distribution and diversity of T. gondii genotypes in other South American 498 countries”. 

Discussion: What are the possible explanation(s) for the results described above such as the differences between the foreign municipalities bordering Brazil and the North of Brazil ?

2.4- A discussion of the strengths and limitations of the study needs to be included.
---

## [Editor Report · Decision Letter 1]

27 Nov 2024

Dear Dr. Pena,

We are pleased to inform you that your manuscript '**A population study of *Toxoplasma gondii* in the Amazon region expands current knowledge of the genetic diversity in South America**' has been provisionally accepted for publication in PLOS Neglected Tropical Diseases.

Best regards,

Hamed Kalani

Academic Editor

Hira Nakhasi

Section Editor

Shaden Kamhawi

co-Editor-in-Chief

Paul Brindley

co-Editor-in-Chief

---

## [Editor Report · Acceptance letter]

Dear Dr. Pena,

We are delighted to inform you that your manuscript, "**A population study of *Toxoplasma gondii* in the Amazon region expands current knowledge of the genetic diversity in South America**," has been formally accepted for publication in PLOS Neglected Tropical Diseases.

Best regards,

Shaden Kamhawi

co-Editor-in-Chief

Paul Brindley

co-Editor-in-Chief
